# CausalX: A Unified and Causally-Interpretable Plug-and-Play Model for Multi-modal Spatio-Temporal Forecasting

**Shiqi Zhang** [1 2]  **Pan Mu** [1 2]  **Hanting Yan** [1]  **Yuchao Zhu** [1]  **Jinglin Zhang** [3]  **Cong Bai** [1 2]

## Abstract

Multi-modal spatio-temporal forecasting underpins many real-world applications but remains challenging due to the complex and evolving interactions across modalities and time steps. Moreover, the lack of interpretability in existing models limits their reliability in safety-critical scenarios. In this paper, we present **CausalX**, a unified and causally interpretable plug-and-play model for multi-modal spatio-temporal forecasting. CausalX achieves interpretability by learning a dynamic causal-inspired graph across modalities and time, whose edge weights quantify causal attribution strength, and are further refined by a diffusion-based generative process guided by structural priors. To overcome the absence of ground-truth causal structures, CausalX aggregates multi-source constraints from causal analysis techniques and a variational autoencoder, spanning predictive, temporal, interventional, and generative aspects to jointly learn a more comprehensive causal-inspired graph. Extensive experiments on real-world forecasting tasks, including pedestrian trajectory prediction and tropical cyclone forecasting, demonstrate that CausalX achieves superior accuracy while producing interpretable causal-inspired graphs. CausalX is modular, architecture-agnostic, and generalizable, offering a new perspective on bridging causal inference and spatio-temporal forecasting.

## 1. Introduction

Multi-modal spatio-temporal forecasting plays a critical role in various fields such as healthcare, environmental monitoring, and autonomous systems (Deng et al., 2024; Bodele et al., 2024; Wang et al., 2024b). By leveraging complementary information from heterogeneous modalities, including images, text, and sensor data, these models can achieve more accurate and robust predictions (Baltrušaitis et al., 2018).

However, due to complex and dynamically evolving interactions across modalities and over time steps, multi-modal spatio-temporal forecasting remains highly challenging (Deng et al., 2024; Long et al., 2025). The high uncertainty, heterogeneity, and dimensionality of multi-modal data further complicate modeling and hinder interpretability, making it difficult to produce clear and human-understandable explanations (Theissler et al., 2022).

In high-stakes, decision-critical domains such as pedestrian trajectory prediction (Bae et al., 2023; 2024) and tropical cyclone forecasting (Zhang et al., 2025; Huang et al., 2023; 2025), this lack of interpretability is particularly detrimental. When models operate as black boxes, even accurate forecasts may fail to gain user trust in decision-critical scenarios (Medina et al., 2024). Therefore, learning to capture complex, dynamically evolving multi-modal dependencies while providing structured and reliable causal explanations remains an open and pressing challenge (Xia et al., 2023). Existing interpretability and causal modeling approaches typically rely on task-specific assumptions or post-hoc analysis (Jiao et al., 2024; Selvaraju et al., 2017; Schwab & Karlen, 2019), which limits their generality and reliability in complex multi-modal forecasting settings.

Another fundamental challenge lies in the absence of ground-truth causal structures (Faller et al., 2024; Cheng et al., 2022), making it difficult to learn faithful and comprehensive causal dependencies without explicit causal supervision. In complex multi-modal spatio-temporal forecasting settings, explicit causal supervision is usually unavailable, making it generally infeasible to recover a uniquely identifiable ground-truth causal structure. This challenge is also recognized in spatio-temporal causal inference: Christiansen et al. (Christiansen et al., 2022) provide a formal

[1]College of Computer Science and Technology, Zhejiang University of Technology, Hangzhou, China [2]Zhejiang Key Laboratory of Visual Information Intelligent Processing, Hangzhou, China [3]School of Control Science and Engineering, Shandong University, Jinan, China. Correspondence to: Pan Mu <panmu@zjut.edu.cn>.

*Proceedings of the 43$^{rd}$ International Conference on Machine Learning*, Seoul, South Korea. PMLR 306, 2026. Copyright 2026 by the author(s).

causal framework for spatio-temporal processes, while noting that the full causal structure is often difficult to specify; Li et al. (Li et al., 2025) formulate a spatio-temporal counterfactual estimation problem and study the identifiability of estimation targets under assumptions, rather than recovering a unique ground-truth causal graph. Motivated by this perspective, we treat the learned graph as a *causal-inspired* and proxy-guided relational structure, rather than a formally identifiable causal graph.

To address these challenges, we propose **CausalX**, a unified and causally interpretable plug-and-play model for multimodal spatio-temporal forecasting. Rather than fixing a predefined structure, CausalX models cross-modal and temporal interactions by constructing dynamic causal-inspired graphs that span modalities and time steps. It learns instance-specific causal structures, enabling not only improved predictive performance but also causal explanations that reveal which modalities and time steps contribute most to each outcome.

Moreover, a single causal signal is often insufficient to reliably characterize such complex dependencies, since different signals capture different aspects of the underlying relational structure. Therefore, CausalX introduces a general and modular causal inference mechanism that integrates multiple complementary causal signals. It incorporates predictive, temporal, interventional, and generative constraints derived from diverse causal analysis techniques, including *Granger* causality (Granger, 1969), time-delayed mutual information (*TDMI*) (Kraskov et al., 2004), *do-calculus* (Pearl, 1994), and a variational autoencoder (*VAE*), to jointly guide causal graph learning.

Starting from an initial causal graph inferred by a graph neural network, whose edge weights serve as causal attributions, the graph is jointly supervised by the multiple causal constraints introduced above. Given that the true causal structure is still unknown and uncertain, CausalX regards the unexplained component of the causal graph as a residual and performs noise-driven iterative denoising to recover a more reliable graph, guided by a coarse prior graph that provides weak structural cues, in order to capture unobserved influences, model mismatch, and noise.

We evaluate CausalX on two representative real-world forecasting tasks: tropical cyclone forecasting and pedestrian trajectory prediction. The former is a typical multi-modal and multi-task forecasting, while the latter is a multi-agent problem dominated by social interactions. Both are high-stakes, decision-critical settings, jointly highlighting the breadth of the method. Extensive experiments demonstrate that CausalX not only achieves superior predictive accuracy but also produces interpretable causal-inspired graphs that reveal key cross-modal and temporal relationships. Furthermore, its modular and architecture-agnostic design ensures strong generalizability, offering a unified perspective for integrating causal interpretability into spatio-temporal forecasting. The main contributions are summarized as follows:

- We propose CausalX, a unified and causally interpretable model for multi-modal spatio-temporal forecasting that explicitly models evolving dependencies across modalities and time steps, providing structured causal-inspired explanations.

- A prior-guided diffusion mechanism is introduced to refine causal-inspired graphs, and multi-source causal constraints that capture predictive, temporal, interventional, and generative dependencies are incorporated to jointly enable a more comprehensive causal-inspired graph.

- CausalX is architecture-agnostic and plug-and-play, allowing seamless integration into existing spatio-temporal forecasting models across diverse tasks and network backbones.

- Extensive experiments on representative high-stakes forecasting tasks, including tropical cyclone and pedestrian trajectory forecasting, demonstrate that CausalX consistently improves predictive accuracy, generalization, and interpretability.

**Conflict of Interest Disclosure.** The authors declare no financial conflicts of interest.

## 2. Preliminary

**Spatio-temporal time series forecasting** aims to predict future values of variables from their past observations, where the variables evolve over time and are associated with spatial locations, movements, interactions, or contextual fields. In the basic form considered in this work, the core target can be formulated as a *1D-to-1D* sequential prediction task, where both input and target are temporal sequences, such as predicting future trajectories from historical ones. Let the historical input be $\mathcal{H} = \{\mathbf{x}_{\text{target}}\}_{t=1}^{T}$, where $\mathbf{x}_{\text{target}}$ is the target value at time step $t$. The goal is to predict future values:

$$\{\mathbf{y}_{\text{target}}\}_{t=T+1}^{T+\tau} = \text{Predictor}\left(\{\mathbf{x}_{\text{target}}\}_{t=1}^{T}\right) \quad (1)$$

**Multi-modal spatio-temporal forecasting** extends this setting by incorporating additional modalities to aid 1D target prediction. The model leverages **(i)** 1D historical values of the target variables, **(ii)** 1D auxiliary variables, and **(iii)** 2D auxiliary inputs such as image-like data. Here, "1D" denotes scalar data. Formally, the model takes $\mathcal{H} = \{\mathbf{x}_{\text{target}}, \mathbf{x}_{\text{aux-1D}}, \mathbf{x}_{\text{aux-2D}}\}_{t=1}^{T}$ as input, and predicts future values of the target variables. CausalX is designed for

the general case where both 1D and 2D auxiliary data are available, but can be easily adapted to single-modal forecasting by omitting auxiliary inputs. This formulation enables broad applicability.

## 3. Methodology

**Overview.** CausalX is a plug-and-play causal-inspired graph learning model for multi-modal spatio-temporal forecasting. Given multi-modal historical inputs, a dynamic graph is built, and is subsequently refined via diffusion under multi-source causal supervision. As shown in Fig. 1, CausalX consists of two stages: **(1) Multi-source causal constraints integration.** An initial graph is derived from historical data as a starting point and regularized by incorporating multiple causal analyses, including *Granger* causality, time-delayed mutual information (*TDMI*), *do-calculus*, and a variational autoencoder (*VAE*), as complementary constraints that capture predictive, temporal, interventional (counterfactual), and generative aspects of causality. **(2) Causal graph refinement via Diffusion.** The graph is further denoised and refined through a diffusion process to reduce residual uncertainty, guided by domain priors.

CausalX outputs (i) a refined **causal-inspired graph** and (ii) its **graph representation**. Depending on the decoder design, one can integrate either the structural graph or the feature representation, enabling easy use of CausalX in diverse backbones.

### 3.1. Multi-Source Causal Constraints Integration

A key challenge in causal inference is the lack of ground-truth causal labels; therefore, we start from a data-driven graph and refine it using multi-source causal constraints.

Given multi-modal spatio-temporal inputs, a fully connected temporal graph is built and initial causal representations are obtained through a GNN encoder. Let $\mathcal{H} = \{\mathbf{x}_{\text{target}}, \mathbf{x}_{\text{aux-1D}}, \mathbf{x}_{\text{aux-2D}}\}_{t=1}^{T}$, where $\mathbf{x}_{\text{target}} \in \mathbb{R}^{B \times T \times N_1}$, $\mathbf{x}_{\text{aux-1D}} \in \mathbb{R}^{B \times T \times N_2}$, and $\mathbf{x}_{\text{aux-2D}} \in \mathbb{R}^{B \times T \times C \times H \times W}$. Each variable (or channel) is treated as a node, yielding $N = N_1 + N_2 + N_3$ nodes, where $N_3 = C$. B is the batch size. Over $T$ historical steps, the total node number is $N_{\text{all}} = N \times T$. A shared *Graph Encoder* extracts node features:
$$\mathbf{F} = f_{\text{enc}}(\mathcal{H}) \in \mathbb{R}^{B \times (TN) \times d} \tag{2}$$
where $d$ is the hidden dimension. As shown in Fig. 7, node features $\mathbf{F}$ are obtained by encoding 1D variables with GRUs and 2D inputs with a CNN-based encoder. A fully connected graph $G = (V, E)$ is then constructed, where each node $v \in V$ corresponds to a feature in $\mathbf{F}$ and each edge $(i, j) \in E$ represents a possible causal link. A Graph Attention Network (GAT) models dependencies:
$$\mathbf{Z}, \mathbf{edge}, \boldsymbol{\alpha} = \text{GAT}(\mathbf{F}, E) \tag{3}$$

where $\mathbf{Z} \in \mathbb{R}^{B \times (TN) \times d}$ is the updated node embedding, **edge** stores the edge index pairs in $E$, and $\boldsymbol{\alpha} \in [0, 1]$ denotes the corresponding attention weights representing causal strength. $\boldsymbol{\alpha}$ contains relationships across time steps, different modalities and different tasks. The initial **C**ausal **G**raph is defined as $\mathcal{CG}_0 = (\mathbf{edge}, \boldsymbol{\alpha})$, which can be easily mapped to a matrix form. A graph-level embedding is obtained via mean pooling over all nodes and time steps:
$$\mathbf{g} = \text{MeanPool}(\mathbf{Z}) \in \mathbb{R}^{B \times d} \tag{4}$$

This construction yields node embeddings $\mathbf{Z}$, a global graph representation $\mathbf{g}$, and an attention-based graph $\mathcal{CG}_0$. However, as it is learned without causal labels, $\mathcal{CG}_0$ may include spurious or incomplete relations. To enhance its reliability, four complementary causal constraints are introduced to cover different aspects of causal reasoning. Specifically, ***Granger* causality** and ***do-calculus*** capture predictive and counterfactual relations, while **time-delayed mutual information (*TDMI*)** models temporal lags, and a **Variational Autoencoder (*VAE*)** addresses nonlinear dependencies through generative reconstruction. Together, these constraints jointly regularize $\mathcal{CG}_0$ from predictive, interventional, temporal, and generative perspectives. Each technique yields a supervision graph.

#### 3.1.1. GRANGER CAUSALITY

We extend *Granger* causality to multi-modal settings to measure *predictive dependencies* among all variables without directly using ground-truth future values. To preserve semantic interpretability and avoid architecture-dependent variations, we compute Granger causality directly on observed input variables rather than on intermediate features, which are commonly used in prior works but often lead to model-specific causal patterns. Given environment variables $\mathbf{x}_{\text{env}} \in \mathbb{R}^{B \times T \times N}$ (concatenating target, 1D, and spatially averaged 2D inputs) and target variables $\mathbf{x}_{\text{target}} \in \mathbb{R}^{B \times T \times N_1}$, we compute Granger scores between each environment variable node $e \in \{1, \ldots, N\}$ and target node $tar \in \{1, \ldots, N_1\}$. Denote the directed effect $e \rightarrow tar$. For a maximum lag order $L$, the score is

$$\text{Granger}(e \rightarrow tar) = 1 - \frac{1}{B} \sum_{b=1}^{B} \min_{\ell \in [1,L]} p\text{-value}_{b,\ell}^{(e \rightarrow tar)} \tag{5}$$

where $p\text{-value}_{b,\ell}^{(e \rightarrow tar)}$ is the significance level (normalized to $[0, 1]$) of the hypothesis that $e$ does not Granger-cause $tar$ on batch $b$ with lag $\ell$. This yields a sparse variable-level matrix $\mathbb{R}^{N \times N_1}$. To align with the fully connected graph $\mathcal{CG}_0 \in \mathbb{R}^{B \times (TN)^2}$, we broadcast each $(e, tar)$ score in $\text{Granger}(e \rightarrow tar)$ to node pairs $(i, j)$ with $i \in [1, N]$ and $j \in [1, N]$ that correspond to $(e, tar)$ (others set to 0), then broadcast along the temporal and batch axis, forming

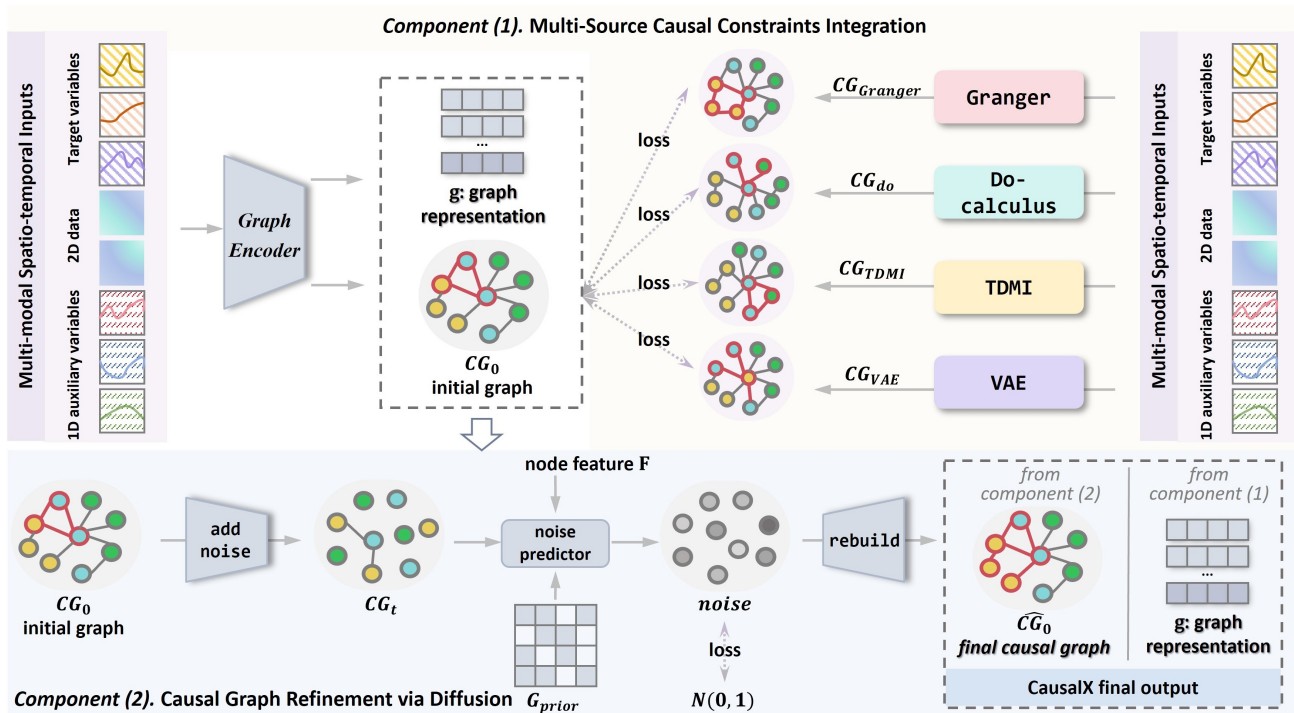

*Figure 1.* Framework of CausalX. It consists of two components: (1) multi-source causal constraints integration, which constructs an initial causal graph $\mathcal{CG}_0$ from multi-modal spatio-temporal inputs via a graph encoder and refines it under complementary causal supervision (*Granger*, *do-calculus*, *TDMI*, and *VAE*); (2) diffusion-based causal graph denoising guided by a prior graph $G_{\text{prior}}$. CausalX outputs a causal-inspired graph $\hat{\mathcal{CG}}_0$ and a graph representation, which can be integrated into the backbone via two integration modes (see Sec. 3.3).

$\mathcal{CG}_{\text{Granger}} \in \mathbb{R}^{B \times (TN)^2}$. Its constraint loss is defined as

$$\mathcal{L}_{\text{GC}} = \frac{1}{E} \sum_{(i,j) \in \mathbf{edge}_{\text{index}}} \left( \alpha_{(i,j)} - \mathcal{CG}_{\text{Granger}}[i,j] \right)^2 \quad (6)$$

which enforces consistency between learned attentions $\alpha_{(i,j)}$ and Granger-derived dependencies.

### 3.1.2. DO-CALCULUS

To capture *interventional (counterfactual) dependencies* beyond predictive relations, we introduce a structural intervention-based constraint inspired by *do-calculus*. Given node features $\mathbf{F} \in \mathbb{R}^{B \times (TN) \times d}$, the edge set $E$, and the node embedding $\mathbf{Z}$, for each node $i$ in all time steps, we simulate an intervention $\text{do}(\mathbf{F}_i)$ by perturbing its feature through mean scaling. Specifically, we replace the feature of node $i$ with the sample-wise mean feature across all nodes:

$$\tilde{\mathbf{F}}_{:,i,:} = \frac{1}{TN} \sum_{m=1}^{TN} \mathbf{F}_{:,m,:}, \quad (7)$$

while keeping all other node features unchanged, i.e., $\tilde{\mathbf{F}}_{:,m,:} = \mathbf{F}_{:,m,:}$ for $m \neq i$. We then rerun the same GAT on the perturbed graph and obtain new node embeddings $\mathbf{Z}^{\text{per}(i)}$. The intervention-inspired effect score from node $i$

to node $j$ is quantified as

$$\mathcal{CG}_{\text{do}}(i \to j) = \frac{1}{d} \sum_{k=1}^{d} \left| \mathbf{Z}_{j,k}^{\text{per}(i)} - \mathbf{Z}_{j,k} \right|. \quad (8)$$

where $k \in \{1, \dots, d\}$. This produces a batch-level causal matrix $\mathcal{CG}_{\text{do}} \in \mathbb{R}^{B \times (TN)^2}$. To align learned attentions $\alpha_{(i,j)}$ with intervention-based effects, the loss is defined as:

$$\mathcal{L}_{\text{do}} = \frac{1}{E} \sum_{(i,j) \in \mathbf{edge}_{\text{index}}} \left( \alpha_{(i,j)} - \mathcal{CG}_{\text{do}}[i,j] \right)^2 \quad (9)$$

### 3.1.3. TIME-DELAYED MUTUAL INFORMATION (TDMI)

While *Granger* and *do-calculus* capture predictive and interventional relations, they do not explicitly quantify *time-lagged (temporal) dependencies*. To model these, we introduce a TDMI-based constraint that measures the maximal mutual information between each environment variable $e$ and target variable $tar$ over delays $\tau \in [1, \tau_{\text{max}}]$:

$$\text{TDMI}(e \to tar) = \max_{\tau} \text{MI}(\mathbf{x}_{\text{env}}^{t-\tau}(e), \mathbf{x}_{\text{target}}^{t}(tar)) \quad (10)$$

After normalization to $[0, 1]$, the scores are temporally broadcast to form $\mathcal{CG}_{\text{TDMI}} \in \mathbb{R}^{B \times (TN)^2}$, and used to constrain $\alpha$, encouraging the model to align high-attention

edges with strong temporal dependencies:

$$\mathcal{L}_{\text{TDMI}} = \frac{1}{E} \sum_{(i,j)\in\mathbf{edge}_{\text{index}}} \left( \alpha_{(i,j)} - \mathcal{CG}_{\text{TDMI}}[i,j] \right)^2 \quad (11)$$

### 3.1.4. VARIATIONAL AUTOENCODER (VAE)

To complement the linear assumptions of classical causal inference, we introduce a *generative* constraint based on a Variational Autoencoder (*VAE*). This constraint captures nonlinear cross-node dependencies from a reconstruction perspective, improving robustness in complex multi-modal settings. Given node features $\mathbf{F} \in \mathbb{R}^{B\times(TN)\times d}$ ,where $\mathbf{F}_j := \mathbf{F}[:,j,:] \in \mathbb{R}^{B\times d}$. For each node pair $(i,j)$, we derive a latent variable $\mathbf{z}_i = \mu_i + \epsilon\sigma_i$, $\epsilon \sim \mathcal{N}(0,I)$ and reconstruct $\hat{\mathbf{F}}_j = f_{\text{dec}}(\mathbf{z}_i)$. The causal score is defined as

$$\text{Score}(i\to j) = -\text{MSE}(\hat{\mathbf{F}}_j, \mathbf{F}_j) \quad (12)$$

considering only valid directions with $t_i < t_j$. After normalization, the resulting matrix $\mathcal{CG}_{\text{VAE}}$ supervises the learned attention via

$$\mathcal{L}_{\text{VAE}} = \frac{1}{E} \sum_{(i,j)\in\mathbf{edge}_{\text{index}}} \left( \alpha_{(i,j)} - \mathcal{CG}_{\text{VAE}}[i,j] \right)^2 \quad (13)$$

### 3.2. Causal Graph Refinement via Diffusion

Although multiple causal constraints have been introduced in Sec.3.1, the estimated graph $\mathcal{CG}_0$ may still miss subtle or uncertain relations, as ground-truth causal structures are unavailable in real-world data. We model these residuals through a conditional diffusion-based graph denoising process, which iteratively refines the causal structure using node features and prior knowledge.

**Diffusion process.** Following the DDPM formulation, the graph is gradually perturbed by Gaussian noise:

$$\mathcal{CG}_t = \sqrt{\bar{\alpha}_t}\,\mathcal{CG}_0 + \sqrt{1-\bar{\alpha}_t}\,\epsilon, \quad \epsilon \sim \mathcal{N}(0,I) \quad (14)$$

where $\{\beta_t\}_{t=1}^T$ controls the noise schedule with $\alpha_t = 1-\beta_t$ and $\bar{\alpha}_t = \prod_{s=1}^t \alpha_s$.

**Denoising network.** A fully connected predictor $f_\theta$ estimates the noise $\hat{\epsilon}$ conditioned on the noisy graph $\mathcal{CG}_t$, node feature $\mathbf{F}$, a prior graph $\mathbf{G}_{\text{prior}}$, and timestep $t$.

$$\hat{\epsilon} = f_\theta(\text{flatten}(\mathcal{CG}_t), \text{flatten}(\mathbf{F}), \text{flatten}(\mathbf{G}_{\text{prior}}), t) \quad (15)$$

Here, $\mathbf{G}_{\text{prior}} \in \{0,1\}^{N\times N}$ is a hand-crafted binary adjacency encoding only consensus relations. Appendix A.1 provides the construction method. The loss is $\mathcal{L}_{\text{diffusion}} = \text{MSE}(\hat{\epsilon}, \epsilon)$. The refined structure is reconstructed as

$$\hat{\mathcal{CG}}_0 = \frac{1}{\sqrt{\bar{\alpha}_t}}\left( \mathcal{CG}_t - \sqrt{1-\bar{\alpha}_t}\,\hat{\epsilon} \right) \quad (16)$$

here we omit reconstruction loss, as the goal is to recover missing causal components rather than replicate $\mathcal{CG}_0$.

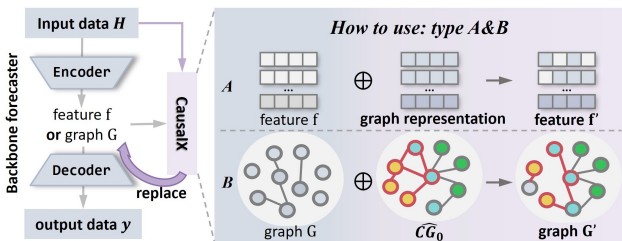

*Figure 2.* Two plug-in modes for CausalX: feature fusion (Type A) and graph fusion (Type B).

The final training objective is formed by an unweighted sum of all terms:

$$\mathcal{L}_{\text{CausalX}} = \mathcal{L}_{\text{GC}} + \mathcal{L}_{\text{do}} + \mathcal{L}_{\text{TDMI}} + \mathcal{L}_{\text{VAE}} + \mathcal{L}_{\text{diffusion}} \quad (17)$$

### 3.3. How to Use CausalX in Different Models

Fig. 2 shows that CausalX can be plugged into an existing backbone depending on whether the decoder consumes **(i)** a *feature* or **(ii)** a *graph* structure. Accordingly, CausalX provides two outputs: **(a)** graph-level representation $\mathbf{g}$ and **(b)** causal-inspired graph $\hat{\mathcal{CG}}_0$, enabling two plug-in strategies:

**Type A: Feature fusion.** For feature-based decoders (e.g., $\text{TCN}_M$ (Huang et al., 2025), MID (Gu et al., 2022)), we concatenate the backbone feature $\mathbf{f} \in \mathbb{R}^{B\times d_{\text{ori}}}$ with CausalX's $\mathbf{g} \in \mathbb{R}^{B\times d}$ and apply a projection to obtain $\mathbf{f}' \in \mathbb{R}^{B\times d_{\text{ori}}}$, which replaces $\mathbf{f}$ as the decoder input.

**Type B: Graph replacement/augmentation.** For graph-structured decoders relying on adjacency (e.g., EigenTrajectory (Bae et al., 2023)), $\hat{\mathcal{CG}}_0$ can replace or be added with the original topology to yield a causally informed graph.

In practice, both forms make CausalX a plug-and-play causal model that can be easily employed across diverse multi-modal spatio-temporal forecasting architectures. Here, plug-and-play refers to modular architectural integration across backbones, please see more in Appendix A.2. We have released the code at https://github.com/Zjut-MultimediaPlus/CausalX.

## 4. Experiments

This section evaluates CausalX by plugging it into representative SOTA backbones on two high-stakes spatio-temporal forecasting domains: tropical cyclone forecasting (multi-modal, multi-task) and pedestrian trajectory prediction (single-modal, multi-agent). The experimental setup and datasets are described first and followed by the quantitative results on both domains and a long-horizon TC study. Component/constraint ablations, a quantitative causal edge faithfulness test, and efficiency/deployment analyses are then reported. Interpretability evidence is finally provided via causal chord diagrams and a multi-agent case study.

## 4.1. Experimental Setup

For fair evaluation, all hyperparameters and protocols are kept identical to those of the corresponding backbone models. A simple unweighted sum of all loss terms is used for optimization, which remains stable in our experiments and avoids additional loss-weight tuning. Appendix A.2 details how to deploy CausalX, showing that CausalX is plug-and-play and easy to integrate across backbones.

**Datasets for pedestrian trajectories.** We follow the standard ETH (Pellegrini et al., 2009)/UCY (Lerner et al., 2007) protocol and report results on five scenes: ETH, Hotel, Univ, Zara1 and Zara2, covering 1,536 pedestrian tracks.

**Dataset for Tropical cyclones.** We use the multi-modal TC dataset from the $TCN_M$ framework (Huang et al., 2025), covering 1,722 Western North Pacific TCs from 1950–2021. It includes TC historical track and intensity records, together with auxiliary inputs (e.g., 2D single-channel geopotential height maps and 1D environmental variables) that are temporally aligned with the track/intensity sequences.

## 4.2. Overall Performance

We evaluate CausalX on two representative spatio-temporal forecasting tasks: pedestrian trajectory (single-modal) and tropical cyclone (TC) forecasting (multi-modal). For each task, we plug CausalX into several SOTA backbones and evaluate the performance. Specifically, pedestrian trajectory forecasting is evaluated using the standard Average Displacement Error (ADE, mean $\ell_2$ error over all predicted time steps) and Final Displacement Error (FDE, $\ell_2$ error at the final time step) under the ETH/UCY protocol, while TC forecasting is assessed using standard trajectory and intensity error metrics at multiple horizons.

**Pedestrian trajectory forecasting.** We consider three recent SOTA backbones: MID (Gu et al., 2022) (CVPR 2022; diffusion-based), EigenTrajectory (Bae et al., 2023) (ICCV 2023; low-rank trajectory descriptors), and SingularTrajectory (Bae et al., 2024) (CVPR 2024; singular-space modeling). Under the ETH/UCY protocol, ADE/FDE are reported. Table 1 shows that integrating CausalX consistently improves all backbones and domains. For example, the average ADE/FDE on MID decreases from $0.248/0.458$ to $0.238/0.420$. A few scene-specific drops appear in ZARA1 or ZARA2, mainly because these scenes are more strongly affected by unobserved environmental constraints that are not explicitly modeled by the original backbones or CausalX; see Appendix B.1 for details. Qualitative examples in Fig. 3 show that the CausalX-augmented predictions (cyan dashed line) are consistently closest to the ground truth (blue solid line).

**Tropical cyclone forecasting.** We study two SOTA multi-modal backbones: $TCN_M$ (Huang et al., 2025) (Nature

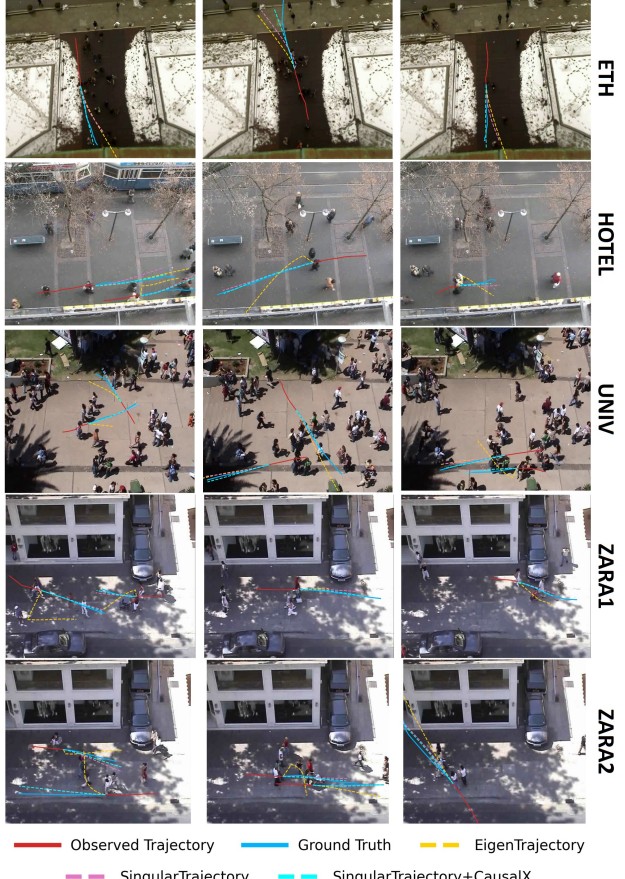

*Figure 3.* Visualization of pedestrian trajectory forecasting. For clarity, the best trajectory among $S = 20$ samples is shown. Right-side labels indicate dataset names.

Communications 2025) and TC-Diffuser (Zhang et al., 2025) (AAAI 2025; diffusion-based). The task jointly predicts future *trajectory* (km), *central pressure* (hPa), and *wind speed* (m/s) at 6–24 h lead times. Table 2 summarizes the results across all horizons/variables. Equipping CausalX yields consistent gains; for $TCN_M$, the 24 h track error decreases from 93.08 to 74.55 km (19.9 km improvement), while TC-Diffuser also benefits. Fig.4 further illustrates accuracy across diverse regimes, including turning tracks, landfall, rapid intensity changes, and near-linear motion, where CausalX-enhanced models remain stable.

Interestingly, $TCN_M$ shows larger gains than TC-Diffuser. TC-Diffuser already imposes shared/specific conditioning that partially disentangles tasks and implicitly captures some cross-task interactions; CausalX therefore brings incremental but still positive benefits by making these relations *explicit* and *time-varying*. By contrast, $TCN_M$ lacks an explicit mechanism for multi-task fusion; inserting CausalX equips it with a unified causal graph over trajectory, pressure, and wind speed, with temporal and cross-modal edges

*Table 1.* Performance when adding CausalX on pedestrian forecasting (ETH/UCY). **Bold**: Better. AVG is the mean over scenes (lower is better). Rows 12/15/18 show the relative change $\Delta$ (%) computed w.r.t. the corresponding backbone.

| Row | Settings | ETH | | HOTEL | | UNIV | | ZARA1 | | ZARA2 | | AVG | |
|---|---|---|---|---|---|---|---|---|---|---|---|---|---|
| | | ADE | FDE | ADE | FDE | ADE | FDE | ADE | FDE | ADE | FDE | ADE | FDE |
| 1 | Social-GAN (Gupta et al., 2018) | 0.810 | 1.520 | 0.720 | 1.610 | 0.600 | 1.260 | 0.340 | 0.690 | 0.420 | 0.840 | 0.580 | 1.180 |
| 2 | STGCNN (Mohamed et al., 2020) | 0.890 | 1.630 | 1.220 | 2.480 | 0.900 | 1.610 | 0.680 | 1.250 | 1.360 | 2.120 | 1.010 | 1.820 |
| 3 | Causal-STGCNN (Chen et al., 2021) | 0.640 | 1.000 | 0.380 | 0.450 | 0.490 | 0.810 | 0.340 | 0.530 | 0.320 | 0.490 | 0.430 | 0.660 |
| 4 | PECNet (Mangalam et al., 2020) | 0.540 | 0.870 | 0.180 | 0.240 | 0.350 | 0.600 | 0.220 | 0.390 | 0.170 | 0.300 | 0.290 | 0.480 |
| 5 | STAR (Yu et al., 2020) | 0.360 | 0.650 | 0.170 | 0.360 | 0.310 | 0.620 | 0.260 | 0.550 | 0.220 | 0.460 | 0.260 | 0.530 |
| 6 | Trajectron++ (Salzmann et al., 2020) | 0.390 | 0.830 | 0.120 | 0.210 | 0.200 | 0.440 | 0.150 | 0.330 | 0.110 | 0.250 | 0.190 | 0.410 |
| 7 | LB-EBM (Pang et al., 2021) | 0.300 | 0.520 | 0.130 | 0.200 | 0.270 | 0.520 | 0.200 | 0.370 | 0.150 | 0.290 | 0.210 | 0.380 |
| 8 | PCCSNET (Sun et al., 2021) | 0.280 | 0.540 | 0.110 | 0.190 | 0.290 | 0.600 | 0.210 | 0.440 | 0.150 | 0.340 | 0.210 | 0.420 |
| 9 | AgentFormer (Yuan et al., 2021) | 1.600 | 2.650 | 1.020 | 1.640 | 1.130 | 1.900 | 1.190 | 2.010 | 1.080 | 1.590 | 1.200 | 1.960 |
| 10 | MID (Gu et al., 2022) | 0.430 | 0.750 | 0.170 | 0.300 | 0.250 | 0.460 | 0.210 | 0.410 | 0.180 | 0.370 | 0.248 | 0.458 |
| 11 | Ours (MID+CausalX) | **0.418** | **0.642** | **0.166** | **0.289** | **0.231** | **0.431** | **0.208** | **0.408** | **0.166** | **0.329** | **0.238** | **0.420** |
| 12 | $\Delta$ (%) | +2.8% | +14.4% | +2.4% | +3.7% | +7.6% | +6.3% | +1.0% | +0.5% | +7.8% | +11.1% | +4.0% | +8.3% |
| 13 | EigenTrajectory (Bae et al., 2023) | 0.375 | 0.613 | 0.130 | 0.209 | 0.250 | 0.448 | 0.205 | 0.356 | 0.162 | 0.273 | 0.224 | 0.380 |
| 14 | Ours (EigenTrajectory+CausalX) | **0.371** | **0.590** | **0.129** | **0.200** | **0.247** | **0.439** | 0.213 | 0.383 | **0.152** | **0.262** | **0.222** | **0.375** |
| 15 | $\Delta$ (%) | +1.1% | +3.8% | +0.8% | +4.3% | +1.2% | +2.0% | -3.9% | -7.6% | +6.2% | +4.0% | +0.9% | +1.3% |
| 16 | SingularTrajectory (Bae et al., 2024) | 0.370 | 0.491 | 0.117 | 0.186 | 0.257 | 0.441 | 0.232 | 0.424 | 0.149 | 0.257 | 0.225 | 0.360 |
| 17 | Ours (SingularTrajectory+CausalX) | **0.361** | **0.480** | **0.116** | **0.183** | **0.248** | **0.429** | **0.219** | **0.402** | 0.155 | 0.269 | **0.220** | **0.353** |
| 18 | $\Delta$ (%) | +2.4% | +2.2% | +0.9% | +1.6% | +3.5% | +2.7% | +5.6% | +5.2% | -4.0% | -4.7% | +2.2% | +1.9% |

*Table 2.* Performance when adding CausalX on tropical cyclone forecasting. Columns report trajectory distance error (km), pressure error (hPa), and wind speed error (m/s) at 6–24 h lead times (lower is better). All metrics are mean squared error (MSE). Rows 1–4: single-task baselines; Rows 5–9: multi-task baselines.

| Row | Settings | Distance (km) | | | | Pressure error (hPa) | | | | Wind speed error (m/s) | | | |
|---|---|---|---|---|---|---|---|---|---|---|---|---|---|
| | | 6h | 12h | 18h | 24h | 6h | 12h | 18h | 24h | 6h | 12h | 18h | 24h |
| 1 | GRU (Cho et al., 2014) | 45.85 | 104.07 | 180.29 | 275.77 | - | - | - | - | - | - | - | - |
| 2 | NMPT (Gao et al., 2018) | 44.10 | 101.72 | 177.06 | 270.91 | - | - | - | - | - | - | - | - |
| 3 | DLM (Pan et al., 2019) | - | - | - | - | - | - | - | - | 1.09 | 1.85 | 2.48 | 3.04 |
| 4 | TCIF-fusion (Wang et al., 2024a) | - | - | - | - | - | - | - | - | - | - | - | 3.56 |
| 5 | SGAN (Gupta et al., 2018) | 28.88 | 61.75 | 98.74 | 140.61 | 1.91 | 3.12 | 4.20 | 5.12 | 1.05 | 1.69 | 2.28 | 2.81 |
| 6 | GBRNN (Alemany et al., 2019) | 29.93 | 65.06 | 105.74 | 152.06 | - | - | - | - | 1.16 | 1.89 | 2.52 | 3.10 |
| 7 | MMSTN (Huang et al., 2022) | 27.57 | 59.09 | 96.54 | 139.19 | 1.69 | 2.86 | 3.94 | 4.74 | 0.95 | 1.52 | 2.10 | 2.55 |
| 8 | TAM-CL (Li et al., 2024) | - | - | - | 155.04 | - | - | - | - | - | - | - | 4.12 |
| 9 | CMO (CMO, 2019) | 37.08 | 52.93 | 60.69 | 75.49 | 2.67 | 4.30 | 5.04 | 6.31 | 2.29 | 3.45 | 2.75 | 5.00 |
| 10 | $\text{TCN}_M$ (Huang et al., 2025) | 23.14 | 43.37 | 67.09 | 93.08 | 1.37 | 2.04 | 2.66 | 3.29 | 0.73 | 1.17 | 1.55 | 1.86 |
| 11 | Ours ($\text{TCN}_M$+CausalX) | **19.55** | **34.81** | **52.82** | **74.55** | **1.36** | **2.03** | **2.64** | **3.22** | **0.70** | **1.14** | **1.47** | **1.84** |
| 12 | $\Delta$ (%) | +15.5% | +19.7% | +21.3% | +19.9% | +0.7% | +0.5% | +0.8% | +2.1% | +4.1% | +2.6% | +5.2% | +1.1% |
| 13 | TC-Diffuser (Zhang et al., 2025) | 19.90 | 20.74 | 41.65 | 77.35 | 1.22 | 0.79 | 1.71 | 2.59 | 0.77 | 0.41 | 0.89 | 1.34 |
| 14 | Ours (TC-Diffuser+CausalX) | **18.88** | **20.38** | **40.21** | **73.76** | 1.23 | **0.74** | **1.61** | **2.39** | **0.76** | **0.40** | **0.86** | **1.22** |
| 15 | $\Delta$ (%) | +5.1% | +1.7% | +3.5% | +4.6% | -0.8% | +6.3% | +5.8% | +7.7% | +1.3% | +2.4% | +3.4% | +9.0% |

learned dynamically, yielding larger improvements. These observations support two key conclusions: **(i)** CausalX effectively captures causal dependencies not only across modalities and time steps, but also across tasks; **(ii)** such dependencies are crucial for accurate multi-task forecasting.

**Longer-term prediction.** Motivated by the strong 24h gains, particularly on $\text{TCN}_M$, we further assess long-range TC trajectory forecasting (up to 72h) to reflect operational needs for hazard preparedness. Results at 72h are reported only for $\text{TCN}_M$, as other backbones expose 24 h settings. Comparisons include TAM-CL (Li et al., 2024) (2024; long-range) and MMSTN (Huang et al., 2022) (GRL 2022, a strong geoscience baseline), and classical baselines. As shown in Table 6, CausalX's gains *increase* with horizon: $\text{TCN}_M$+CausalX cuts the 72h error by 92.35km and even

surpasses TAM-CL, while its dynamic causal fusion preserves informative despite long-horizon challenges.

**Ablation study.** Table 3 and Table 5 show that CausalX benefits from both multi-source causal constraints (CauCon) and diffusion-based refinement (RefDif). Importantly, ablating individual causal constraints within CauCon consistently degrades performance, indicating that each constraint provides complementary supervision. Among them, removing *Granger* leads to the largest drop on $\text{TCN}_M$, while removing *do-calculus* is most detrimental on SingularTrajectory, suggesting these two constraints are particularly influential. Finally, enabling RefDif further improves results, and the full CausalX achieves the best performance.

**Quantitative causal edge faithfulness and stability.**

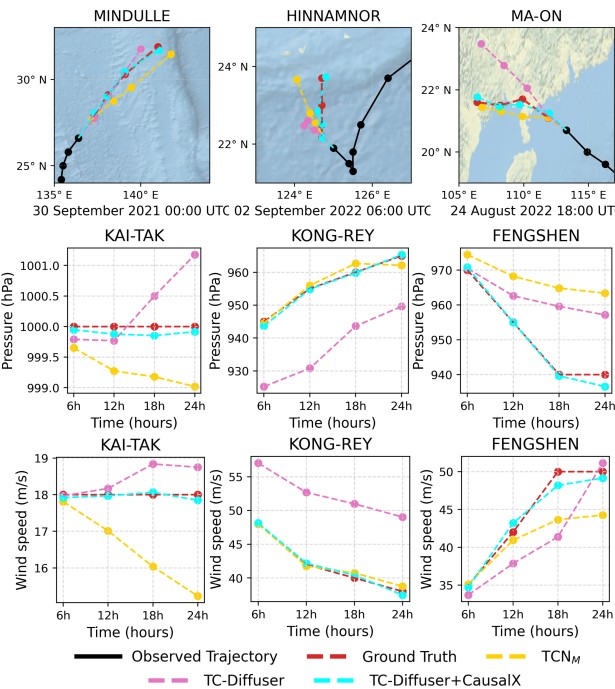

*Figure 4.* Visualization of tropical cyclone forecasting. For clarity, the best trajectory among $S = 6$ samples is shown. Panel titles indicate tropical cyclone names.

*Table 3.* Ablation of CausalX components. For $\text{TCN}_M$, we report Distance (km), Pressure (hPa), and Wind speed (m/s). For SingularTrajectory, we report ADE/FDE on UNIV.

| Setting | CauCon | | | | RefDif | $\text{TCN}_M$ | SingularTrajectory |
| | Granger | TDMI | Do | VAE | | Dist/Pres/Wind | ADE/FDE |
|---|---|---|---|---|---|---|---|
| None (Backbone) | | | | | | 226.68/9.36/5.31 | 0.257/0.441 |
| w/o Granger | | ✓ | ✓ | ✓ | | 203.47/9.52/5.24 | 0.252/0.445 |
| w/o TDMI | ✓ | | ✓ | ✓ | | 206.42/9.54/5.25 | 0.252/0.448 |
| w/o Do | ✓ | ✓ | | ✓ | | 207.24/9.54/5.24 | 0.253/0.446 |
| w/o VAE | ✓ | ✓ | ✓ | | | 200.78/9.50/5.23 | 0.252/0.444 |
| w/o RefDif | ✓ | ✓ | ✓ | ✓ | | 189.43/9.44/5.21 | 0.250/0.439 |
| CausalX | ✓ | ✓ | ✓ | ✓ | ✓ | **181.74/9.25/5.14** | **0.248/0.429** |

level *do-calculus* (multiple GAT passes per batch), while *Granger/TDMI* graphs are precomputed offline. At inference, all auxiliary modules are disabled and CausalX reduces to a single learned graph representation fused into the decoder input; accordingly, Table 4 reports only marginal latency changes and a small parameter increase, confirming negligible deployment overhead.

Overall, these results demonstrate that: **(i)** CausalX is model-agnostic and can be seamlessly integrated into various backbones with little deployment overhead and faster convergence; **(ii)** performance gains are consistent across different datasets, horizons and tasks; **(iii)** each causal constraint plays a vital role and causal-inspired graph is predictively used.

### 4.3. Causal Graph and Interpretability Analysis

**(I): multi-task structure across domains.** We visualize causal chord diagrams to identify, from the target's perspective, which modalities, time lags, and nodes most strongly influence the prediction, across two domains (TC and pedestrians) and three backbones (TC-Diffuser, $\text{TCN}_M$, SingularTrajectory); more models appear in Appendix. C. For Fig. 5a (TC-Diffuser+CausalX), we use $T=8$ historical steps and three modality groups with sizes $N_1=4$, $N_2=1$, $N_3=5$ (10 variables per step; 80 historical nodes in total).

CausalX assigns high strengths to physically plausible recent-state predictors. In **TC-Diffuser+CausalX** (Fig. 5a), longitude/latitude are mainly informed by recent positions (longitude_3, latitude_3) and the heading cue FutDir24h_4; pressure is most strongly associated with intenCls_4 and recent pressure_3; and wind speed is dominated by the recent-speed term windSpd_5, consistent with standard TC forecasting heuristics. $\text{TCN}_M$**+CausalX** (Fig. 5b) exhibits similar dependencies and further identifies month_1→longitude, which aligns with seasonal dynamics in the Pacific, where TC activity and the steering subtropical high peak in summer, indicating that CausalX can exploit coarse seasonal priors. **SingularTrajectory+CausalX** (Fig.5c) produces a simpler pattern over two variables (Lon/Lat). Unlike TC, multiple agents (pedestrians) interact at the same time; we therefore aggregate graphs at a median crowd size ($num$=6).

Intervention-style evidence is provided via inference-time edge removal and retention to validate the causal-inspired graph: Fig. 9 shows that removing the top-ranked edges causes larger degradation than random removal, while keeping only the top-ranked edges results in the smallest degradation, suggesting that high-strength edges are both necessary and sufficient for preserving prediction performance (see Appendix. B.3). We further test graph stability by retraining SingularTrajectory+CausalX under different random seeds and independently resampled training splits, obtaining 8 learned graphs. Spearman rank correlation is computed over the flattened edge-weight rankings to measure global ranking consistency, while Top-30% Jaccard overlap compares the sets of strongest 30% edges to measure salient-edge consistency. Across all 28 graph pairs, the mean Spearman correlation is $0.87 \pm 0.06$, and the Top-30% Jaccard overlap is $0.74 \pm 0.07$, indicating that both the overall edge ranking and the salient-edge set are largely preserved and demonstrating strong graph stability.

**Efficiency and Deployment Overhead.** CausalX remains lightweight in deployment and often *reduces time-to-target* in practice. Although running the same epoch budget can slightly increase wall-clock training time, Fig. 8 shows that CausalX reaches (and often surpasses) the backbone's best validation performance noticeably earlier across all five backbones, indicating faster and better convergence.

The extra training cost mainly comes from the feature-

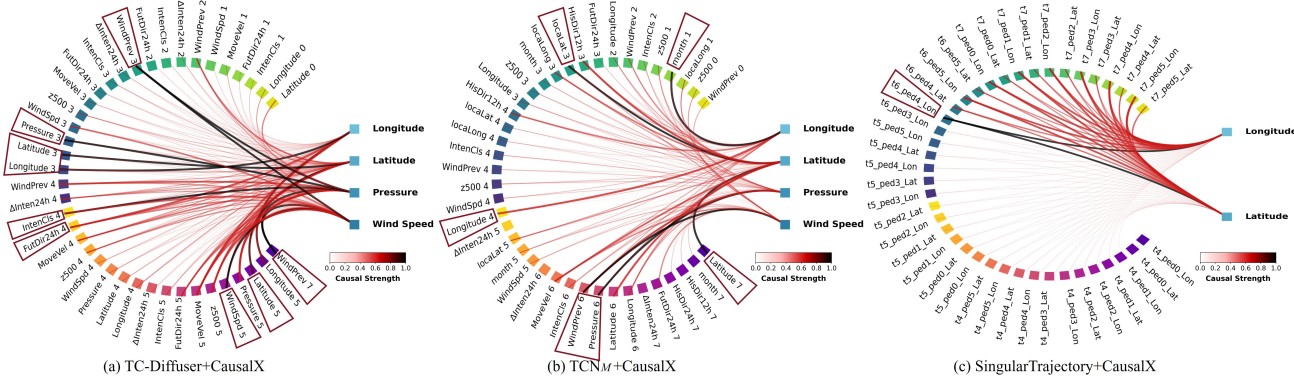

*Figure 5.* Causal chord diagrams (top-40 *source nodes*) learned by CausalX. Historical nodes are ranked by their causal effect on the target at $t=T$; we keep the top-40 and draw only edges from these nodes to the target (self-loops and within-historical edges omitted). For (a)(b), Node labels indicate the variable name and time index. For (c), node labels use `time_pedestrian number_name`, averaged over scenes with $n=6$ pedestrians.

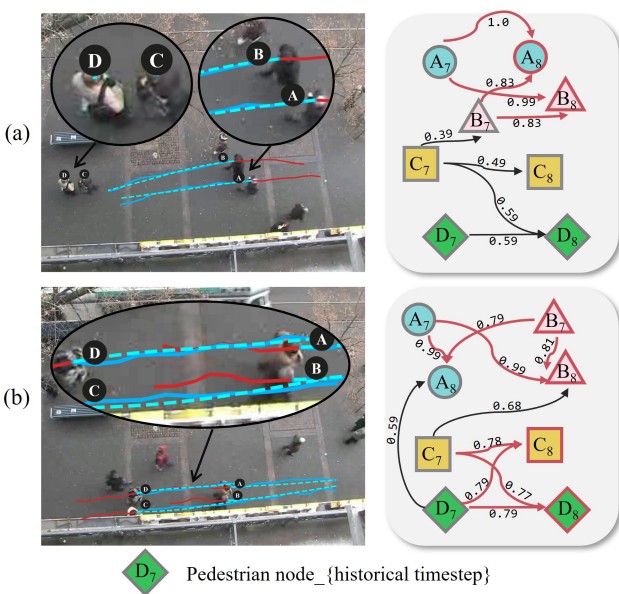

*Figure 6.* Visualization of the causal graph for pedestrian scenarios in the HOTEL dataset when there are 4 pedestrian nodes. Causal strength is shown on edges; strengths $> 0.75$ denote strong causality and are rendered in red, and the target node is highlighted.

Salient edges concentrate at the most recent two lags, and pedestrian 4 often acts as a hub which matches the dataset convention where the central actor in six-agent scenes is indexed as ID 4 and thus interacts with multiple neighbors. Overall, the observed patterns are consistent with these modeling and data-design priors.

**(II) Multi-node causal graph case study.** As a complement to the aggregate statistics, we provide a qualitative example of inter-agent dependencies for pedestrian forecasting (Fig. 6). For clarity, we visualize a four-person scene (agents A–D). In panel (a), agents A and B walk side-by-side with aligned headings and decreasing separation;

correspondingly, the learned graph assigns a strong bidirectional link $A \leftrightarrow B$ along their motion direction, while C and D remain nearly stationary and exhibit weak connections to others. In panel (b), two moving pairs emerge and interact: the graph strengthens within-pair links ($A \leftrightarrow B$, $C \leftrightarrow D$) and increases cross-pair influences relative to panel (a), matching the interaction patterns visible in the frames. Overall, the strongest edges coincide with the interacting agents, suggesting that CausalX captures non-trivial inter-agent dependencies beyond individual motion cues.

**(III) Per-constraint graph visualization.** To make the learned graph more transparent, Fig. 12 reports chord diagrams for each constraint: *Granger*, *do-calculus*, *TDMI*, *VAE*, and the final fused graph. These per-constraint graphs exhibit distinct and complementary patterns: *Granger* and *do-calculus* concentrate on short-lag drivers, *TDMI* emphasizes mid-lag temporal dependencies, and the *VAE* surfaces diverse cross-modal couplings. Their fusion yields a more complete yet physically consistent structure than any single constraint, underscoring that each constraint contributes a non-redundant signal to the final graph.

## 5. Conclusion

We presented CausalX, a unified and causally interpretable plug-and-play model for multi-modal spatio-temporal forecasting. CausalX constructs dynamic causal-inspired graphs to explicitly model cross-modal and temporal dependencies, and refines them through a diffusion-based generative process under multiple complementary causal constraints. Extensive experiments on representative forecasting tasks demonstrate that CausalX not only improves predictive accuracy but also produces interpretable and visualizable causal structures. The modular, architecture-agnostic design of CausalX facilitates extension to a wide range of spatiotemporal domains.

## Acknowledgements

This work is partially supported by the National Natural Science Foundation of China under Grant No. U24A20221, the Zhejiang Provincial Natural Science Foundation of China under Grant No. LRG25F020002, the Zhejiang Provincial Basic Research Program ("Xinmiao" Project), China, under Grant No. 2026XMGD016, and the Taishan Scholars Program under Grant No. tstp20250708.

## Impact Statement

This work aims to improve both the accuracy and reliability of multi-modal spatio-temporal forecasting in decision-critical scenarios by enabling models to provide human-understandable causal explanations alongside strong predictive performance. Such interpretability can support safer and more accountable deployment of predictive models in high-stakes settings, helping practitioners better assess when predictions should be trusted and diagnose potential failure modes. Beyond the tasks studied, the proposed CausalX is general and can be easily extended to other domains that require robust forecasting and transparent reasoning, with broader potential benefits for public safety and risk-aware decision support in complex real-world systems.

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

## Appendix Overview

This appendix is organized into three parts. **Deployment details** (Sec. A) reports the compute setup and implementation details, and summarizes efficiency and deployment overhead (added parameters, runtime/memory) with a two-row-per-backbone table; it further analyzes convergence behavior under wall-clock training time and specifies the construction of the prior graph $G_{\text{prior}}$ used in refinement. **Additional Experiments** (Sec. B) extends ablations of *CauCon* and *RefDif* across tropical-cyclone and pedestrian forecasting backbones, including leave-one-out studies for each causal constraint and an inference-time graph-editing faithfulness test (Top-$k$ remove, random remove, and Top-$k$ keep) to verify that learned edge strengths are predictively utilized. Table 6 provides longer-horizon prediction results to evaluate CausalX's performance in long-term forecasting. **Additional Visualization** (Sec. C) presents interpretability results with chord diagrams for each supervision signal (*Granger*, *do-calculus*, *TDMI*, *VAE*) and for the fused graph, demonstrating complementary emphases and thereby the necessity of each supervision source.

## A. Deployment details

To ensure a fair and consistent comparison, all experiments and efficiency measurements were conducted on a single NVIDIA A6000 GPU.

**Implementation details**

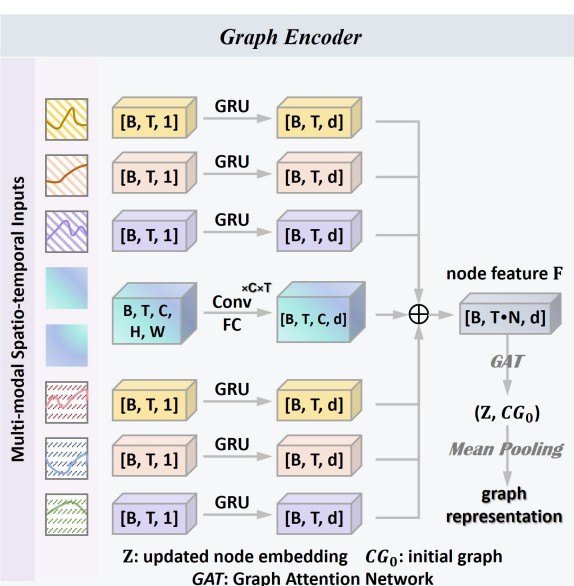

*Figure 7.* Graph encoder details. We encode each 1D variable using an independent GRU and encode 2D inputs with a CNN followed by a projection to dimension $d$. The resulting embeddings are concatenated to form node features $\mathbf{F} \in \mathbb{R}^{B \times (TN) \times d}$, which are fed into a GAT to produce updated node embeddings $\mathbf{Z}$ and an initial attention-based graph $\mathcal{CG}_0$. Mean pooling over nodes yields a graph-level representation used by downstream forecasters.

### A.1. Construction of the prior graph $\mathbf{G}_{\text{prior}}$ in causal graph refinement

**Definition.** The prior graph $\mathbf{G}_{\text{prior}} \in \{0, 1\}^{N \times N}$ is a fixed binary adjacency matrix that encodes coarse, widely accepted domain knowledge about which variables are expected to be directly related. Each entry $\mathbf{G}_{\text{prior}}(i, j) = 1$ indicates that variable $v_i$ is considered a plausible driver (or strongly associated factor) of variable $v_j$ according to domain consensus, and $\mathbf{G}_{\text{prior}}(i, j) = 0$ otherwise. The matrix is fixed throughout training and inference.

**Example (tropical cyclone setting).** For a concrete instantiation, consider a minimal variable set $\mathcal{V} = \{\text{WindSpeed}, \text{Pressure}, \text{GPH}\}$, where rows denote *sources/drivers* and columns denote *targets*. Based on common domain consensus, we include (i) the strong coupling between pressure and wind speed, and (ii) geopotential height (GPH) as

a large-scale environmental factor connected to TC intensity variables. One example binary prior matrix is:

$$\mathbf{G}_{\text{prior}} = \begin{array}{c} \\ \text{WindSpeed} \\ \text{Pressure} \\ \text{GPH} \end{array} \begin{array}{ccc} \text{WindSpeed} & \text{Pressure} & \text{GPH} \\ \left[ \begin{array}{ccc} 1 & 1 & 0 \\ 1 & 1 & 0 \\ 1 & 1 & 1 \end{array} \right] \end{array}. \tag{18}$$

Here, for example, $\mathbf{G}_{\text{prior}}(\text{P}, \text{WS}) = 1$ encodes the widely accepted relation between pressure and wind intensity, while $\mathbf{G}_{\text{prior}}(\text{GPH}, \text{WS}) = 1$ indicates that GPH is treated as a plausible driver of intensity. All other entries not covered by $\mathcal{R}$ are set to 0.

*Note.* The above example is for illustration; in our experiments we construct $\mathbf{G}_{\text{prior}}$ using the full variable list $\mathcal{V}$. In the appendix, we provide a concrete example of the data input format.

### A.2. How to Deploy CausalX

This part summarizes a practical recipe for integrating CausalX into an existing spatio-temporal forecasting backbone.

**Step 1: Prepare inputs for CausalX.** Given a spatio-temporal time series input, split the inputs by dimensionality into (i) 1D variables and (ii) 2D fields. Concretely, the 1D input is formatted as $\mathbf{X}^{\text{1D}} \in \mathbb{R}^{B \times T \times N_{\text{1D}}}$, and the 2D input is formatted as $\mathbf{X}^{\text{2D}} \in \mathbb{R}^{B \times T \times C \times H \times W}$. CausalX additionally takes a binary prior matrix $\mathbf{G}_{\text{prior}} \in \{0, 1\}^{N \times N}$, where $N = N_{\text{1D}} + C$. It is pre-defined as a 0–1 matrix based on which variables are included, following the construction in Sec. A.1 (and the example therein), and is provided as an extra input alongside $\mathbf{X}^{\text{1D}}$ and $\mathbf{X}^{\text{2D}}$. These tensors are then fed into CausalX, which outputs a graph-level representation $\mathbf{g}$ and a (refined) causal-inspired graph $\hat{\mathcal{CG}}_0$.

**Step 2: Locate the decoder interface in the backbone.** Identify the input to the backbone decoder, which typically is either a feature vector $\mathbf{f}$ or a graph structure. In TC forecasting backbones, this interface is often a feature vector $\mathbf{f} \in \mathbb{R}^{B \times d_{\text{ori}}}$.

**Step 3: Plug in CausalX.** For feature-based decoders, fuse the CausalX representation $\mathbf{g} \in \mathbb{R}^{B \times d}$ with the backbone feature $\mathbf{f}$ using a simple strategy (e.g., concatenation followed by a linear projection), and replace $\mathbf{f}$ with the fused feature $\mathbf{f}'$ as the decoder input. For graph-based decoders, $\hat{\mathcal{CG}}_0$ can be used to replace or be added to the original graph.

**Step 4: Training objective.** Add the CausalX losses to the backbone loss and train the integrated model end-to-end from scratch:

$$\mathcal{L}_{\text{total}} = \mathcal{L}_{\text{backbone}} + \mathcal{L}_{\text{CausalX}}.$$

In practice, the loss terms are combined by a simple unweighted sum, and we did not observe the need for additional loss-weight tuning. Overall, CausalX can be deployed as a plug-and-play model with minimal changes to existing architectures.

**Meaning of plug-and-play.** Here, plug-and-play refers to modular architectural integration across backbones, rather than zero-retraining insertion into an already trained pipeline. When the input configuration changes, CausalX naturally requires re-training because the graph is defined over the input variables; when inputs are aligned, the proxy-graph generation process, including offline Granger/TDMI computations and the prior graph, can be reused, reducing adaptation to lightweight fine-tuning or retraining of the fusion module.

### A.3. Computational efficiency and accelerated convergence

Overall, considering *compute budget*, *convergence speed*, and *inference latency*, CausalX remains efficient in practice. Although the total training time in Table 4 may appear longer, this is mainly because all variants are trained for the *same number of epochs* as their backbones for fairness. In terms of *time-to-target* on the validation/test set, enabling CausalX substantially accelerates optimization and leads to faster and better convergence, as evidenced in Fig. 8. In other words, CausalX typically reaches the backbone's best performance earlier and continues to improve beyond it, indicating that the additional per-epoch cost does not translate to a proportional increase in *effective* training time.

Fig. 8 reports the relative performance versus training time across five backbones in two domains. The x-axis measures wall-clock training time, while the y-axis is normalized by the *best performance achieved by the backbone alone* (reference line at 1.0). Values above 1.0 indicate that the CausalX-enhanced model surpasses the backbone's best result, and higher is better. Across all backbones, CausalX exhibits a consistently steeper improvement trajectory, reaching the backbone's

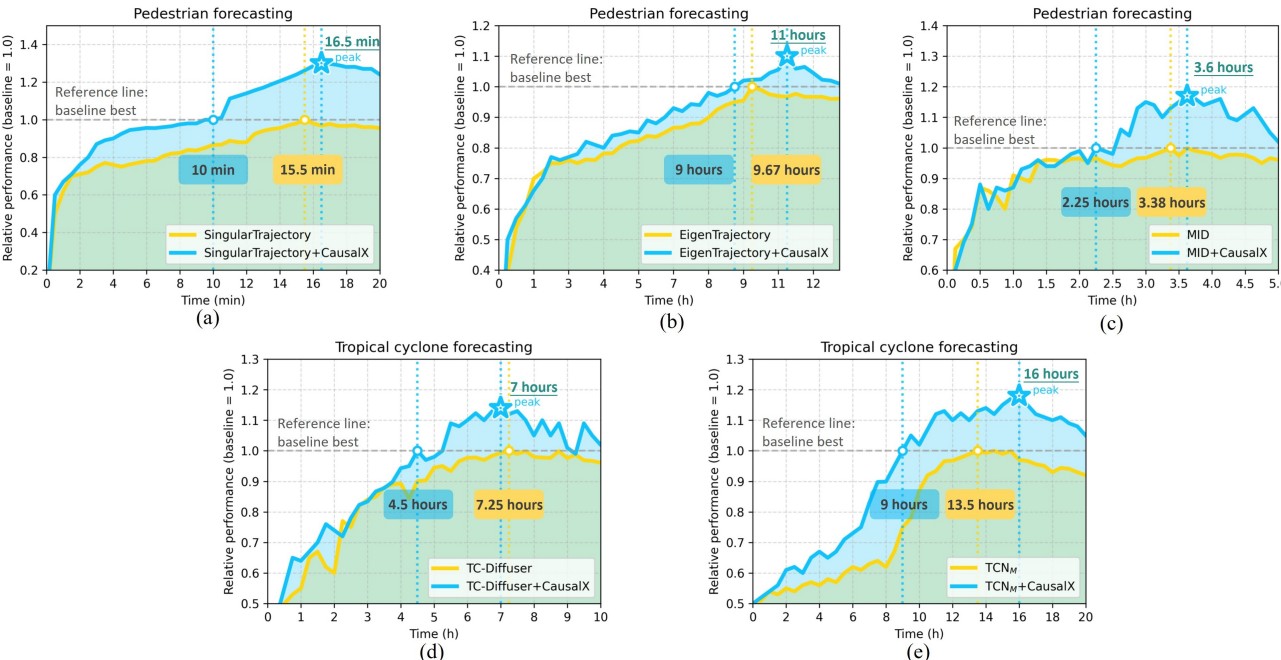

*Figure 8.* **Convergence speed under training time.** Relative performance versus training time for five backbones across pedestrian forecasting (a–c) and tropical cyclone forecasting (d–e). The y-axis is normalized by the backbone-only best performance (reference line at 1.0), so values > 1.0 indicate outperforming the backbone's best result, and higher is better. Dotted vertical lines mark the time when each method first reaches the backbone's best performance (y=1.0), and the star indicates the peak time of the CausalX-enhanced model. Blue (resp. yellow) boxes annotate the training time required for +CausalX (resp. the backbone) to first reach the backbone-best level (y=1.0). Across all backbones, CausalX reaches the backbone's best performance earlier and often surpasses it before the backbone achieves its own peak, demonstrating faster convergence and improved training-time efficiency on validation/test performance.

best performance (y=1.0) earlier. Moreover, even when the CausalX model attains its peak later than the backbone's own peak, it often already exceeds the backbone at the time the backbone reaches its best result, reflecting improved optimization efficiency and generalization.

Meanwhile, the deployment overhead remains minimal. As summarized in Table 4, CausalX increases the model size only slightly (0.2–2.0M depending on the backbone), and the inference-time latency is virtually unchanged (e.g., MID: 0.75s→0.77s; EigenTrajectory: 26ms→29ms; TC-Diffuser: 1.04s→1.05s). The main additional training cost comes from feature-level *do-calculus*, which invokes several GAT passes per batch, while the supervision graphs for *Granger* and *TDMI* are computed offline and loaded with the dataset. Crucially, auxiliary causal modules are disabled at test time, and CausalX reduces to a single learned causal graph fused into the decoder input, making it practical for deployment while delivering consistent performance gains.

**Code.** The full code is available at https://github.com/Zjut-MultimediaPlus/CausalX.

# B. Additional Experiments

## B.1. Scene-Specific Behavior on ZARA

Although CausalX improves the average ADE/FDE of all three pedestrian forecasting backbones, a few scene-specific drops appear in ZARA1 or ZARA2. Compared with more open scenes such as HOTEL or UNIV, the ZARA scenes are more strongly constrained by environmental factors, including buildings, narrow walkable areas, and occasional vehicle interference. These factors can visibly affect pedestrian motion, but they are not explicitly modeled by the tested backbones.

This is important because CausalX uses only the information already available to the original backbone. For a fair comparison, we do not manually introduce additional scene-context inputs, such as maps, obstacle layouts, or vehicle annotations, that are not used by the backbone itself. Therefore, when trajectories are strongly influenced by unobserved scene constraints, the causal-inspired graph learned by CausalX can only organize and refine the available pedestrian-motion

*Table 4.* Efficiency summary when integrating CausalX (two-row per backbone). Perf. denotes the task metric, measured as the sum of all evaluation indicators (e.g., ADE+FDE across ETH/HOTEL/UNIV/ZARA1/ZARA2 for pedestrian forecasting; and for TC forecasting, the sum of all reported metrics across the three targets (trajectory, pressure, and wind speed) over all prediction horizons). $\Delta$ indicates the relative change. Model size is reported in millions of parameters (M). Training and inference times are averaged on the same hardware. Here, h, min, s, and ms denote hours, minutes, seconds, and milliseconds, respectively.

| Task | Backbone | Variant | Perf.↓ | $\Delta$ | Model size (M) | Training / Inference time |
|---|---|---|---|---|---|---|
| **Pedestrian** | MID (Gu et al., 2022) | Baseline | 3.390 | +3.1% | 0.6M | 5h / 0.75s |
| | | +CausalX | 3.288 | | 0.8M | 7h / 0.77s |
| | EigenTrajectory (Bae et al., 2023) | Baseline | 3.021 | +1.2% | 0.07M | 13h / 26ms |
| | | +CausalX | 2.986 | | 0.35M | 16h / 29ms |
| | SingularTrajectory (Bae et al., 2024) | Baseline | 2.924 | +2.2% | 1.9M | 20min / 16ms |
| | | +CausalX | 2.862 | | 2.3M | 45min / 18ms |
| **TC Forecasting** | $\text{TCN}_M$ (Huang et al., 2025) | Baseline | 241.35 | +18.7% | 3.6M | 20h / 0.18s |
| | | +CausalX | 196.13 | | 5.6M | 28h / 0.18s |
| | TC-Diffuser (Zhang et al., 2025) | Baseline | 169.36 | +4.1% | 9.5M | 10h / 1.04s |
| | | +CausalX | 162.44 | | 9.8M | 14h / 1.05s |

*Table 5.* Ablation of CausalX components on all backbones. "CauCon" consists of four multi-source causal constraints; "RefDif" denotes diffusion-based graph refinement. For pedestrian backbones (MID, EigenTrajectory, SingularTrajectory), we report UNIV ADE/FDE. For TC backbones ($\text{TCN}_M$, TC-Diffuser), we report the sum of 6h–24h Distance (km), Pressure (hPa), and Wind speed (m/s). Constraint-level ablations for representative backbones are reported in the main paper (Table 3).

| Setting | CauCon | | | | RefDif | Tropical cyclone (Dist/Pres/Wind) | | Pedestrian (UNIV ADE/FDE) | | |
|---|---|---|---|---|---|---|---|---|---|---|
| | Granger | TDMI | Do | VAE | | $\text{TCN}_M$ | TC-Diffuser | MID | EigenTrajectory | SingularTrajectory |
| None (Backbone) | | | | | | 226.68 / 9.36 / 5.31 | 159.64 / 6.31 / 3.41 | 0.250/0.460 | 0.250/0.448 | 0.257/0.441 |
| w/o Granger | | ✓ | ✓ | ✓ | | 203.47 / 9.52 / 5.24 | 157.60 / 6.30 / 3.40 | 0.241/0.441 | 0.249/0.446 | 0.252/0.445 |
| w/o TDMI | ✓ | | ✓ | ✓ | | 206.42 / 9.54 / 5.25 | 158.02 / 6.32 / 3.42 | 0.240/0.451 | 0.249/0.448 | 0.252/0.448 |
| w/o Do | ✓ | ✓ | | ✓ | | 207.24 / 9.54 / 5.24 | 158.10 / 6.32 / 3.41 | 0.242/0.447 | 0.250/0.446 | 0.253/0.446 |
| w/o VAE | ✓ | ✓ | ✓ | | | 200.78 / 9.50 / 5.23 | 157.39 / 6.30 / 3.41 | 0.239/0.437 | 0.249/0.444 | 0.252/0.444 |
| w/o RefDif | ✓ | ✓ | ✓ | ✓ | | 189.43 / 9.44 / 5.21 | 156.41 / 6.34 / 3.30 | 0.232/0.433 | 0.248/0.441 | 0.250/0.439 |
| CausalX | ✓ | ✓ | ✓ | ✓ | ✓ | **181.74 / 9.25 / 5.14** | **153.23 / 5.97 / 3.24** | **0.231/0.431** | **0.247/0.439** | **0.248/0.429** |

information, rather than compensate for missing environmental variables.

As a result, the improvement in ZARA scenes can be less stable than in more open and interaction-dominated scenes such as HOTEL or UNIV. This explains why CausalX improves the overall average performance across backbones while still showing a few local drops on ZARA1 or ZARA2.

### B.2. Additional ablation analysis

As mentioned in the "Ablation study" section of the main paper, Table 5 reports extended ablation results across five backbones, including two tropical cyclone forecasting backbones ($\text{TCN}_M$, TC-Diffuser) and three pedestrian predictors (MID, EigenTrajectory, SingularTrajectory). We evaluate two components of CausalX: *multi-source causal constraints integration* (CauCon) and *diffusion-based causal graph refinement* (RefDif).

Overall, enabling CauCon consistently improves performance over the corresponding backbones, and further adding RefDif yields additional gains, with the full CausalX achieving the best results in all cases reported. Moreover, ablating any single constraint within CauCon degrades performance, indicating that the four constraints provide complementary supervision.

This trend is consistent across domains and metrics. For tropical cyclone forecasting, both $\text{TCN}_M$ and TC-Diffuser reduce errors in distance, central pressure, and wind speed when progressively enabling CauCon and RefDif. For pedestrian trajectory prediction, all three backbones improve on UNIV ADE/FDE, and the full configuration achieves the lowest errors. Overall, these supplementary results confirm that each component of CausalX plays a distinct and complementary role, and together they form a coherent and effective causal enhancement applicable to a wide range of backbone architectures.

*Table 6.* Performance when adding CausalX on long-term tropical cyclone trajectory forecasting.

| Settings | Distance (km) | | | | |
|---|---|---|---|---|---|
| | 6h | 12h | 24h | 48h | 72h |
| RNN | - | - | 220.09 | 402.85 | 603.22 |
| CNN-LSTM | - | - | 199.33 | 351.67 | 476.29 |
| Conv-LSTM | - | - | 175.89 | 265.41 | 416.64 |
| MMSTN | 27.57 | 59.09 | 139.18 | 336.16 | 544.16 |
| TAM-CL | - | - | 155.04 | 231.45 | 365.60 |
| $\text{TCN}_M$ | 22.98 | 43.83 | 93.75 | 238.46 | 408.35 |
| **Ours ($\text{TCN}_M$ + CausalX)** | **18.48** | **34.60** | **73.07** | **189.80** | **316.00** |

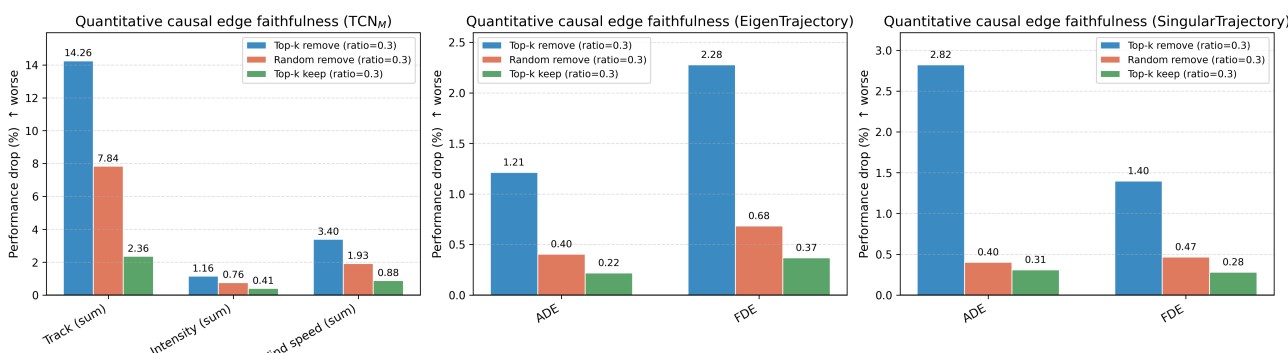

*Figure 9.* **Quantitative causal edge faithfulness under inference-time graph editing.** Bars show the relative performance drop ($\Delta\%$; higher means a larger degradation) with respect to the unedited learned graph. Three editing strategies are compared at the same ratio of $0.3$: **Top-$k$ remove**, which removes edges with the highest learned causal-attribution scores; **Random remove**, which removes the same number of edges uniformly at random; and **Top-$k$ keep**, which keeps only the highest-scoring edges and masks the remaining edges. Results are reported for tropical cyclone forecasting with $\text{TCN}_M$ (Track/Intensity/Wind Speed; summed within each group) and pedestrian forecasting with EigenTrajectory and SingularTrajectory (ADE/FDE). Top-$k$ removal consistently causes the largest degradation, while Top-$k$ keep yields the smallest degradation, indicating that high-strength edges are both important to remove and sufficient to preserve most predictive performance.

### B.3. Causal edge faithfulness via inference-time graph editing.

Figure 9 evaluates whether the learned causal-inspired graph is *functionally used* by the downstream forecasters, rather than being only qualitatively interpretable. The key conclusion is that **high-strength causal edges are both important and sufficient for prediction**: removing them induces the largest degradation, while keeping only them leads to the smallest degradation among all graph-editing strategies.

**Protocol.** We keep each backbone fully trained and fixed, and edit only the inferred causal-inspired graph at inference time. We compare three graph-editing policies with the same ratio of $0.3$. **Top-$k$ remove** removes edges ranked highest by the learned causal-attribution scores. **Random remove** removes the same number of edges uniformly at random. **Top-$k$ keep** keeps only the highest-scoring edges and masks all remaining edges. We report the relative performance drop $\Delta\%$ with respect to the unedited graph, where a larger value indicates larger degradation.

**Evidence.** As shown in Fig. 9, *across all reported metrics for all three backbones*, Top-$k$ removal consistently yields the largest performance drop, while Top-$k$ keep yields the smallest drop. This pattern holds for tropical cyclone forecasting with $\text{TCN}_M$ (aggregated within each target group: Track/Intensity/Wind Speed) and for pedestrian forecasting with **EigenTrajectory** and **SingularTrajectory** (ADE/FDE). The Top-$k$ remove results indicate that high-strength edges are important: when the most influential edges are removed, the model degrades more than under an unstructured perturbation of the same magnitude. The Top-$k$ keep results further provide a sufficiency test: retaining only the strongest edges preserves substantially more predictive performance than random editing. Together, these results support that the learned edge strengths capture predictive dependencies and are functionally used by downstream forecasters.

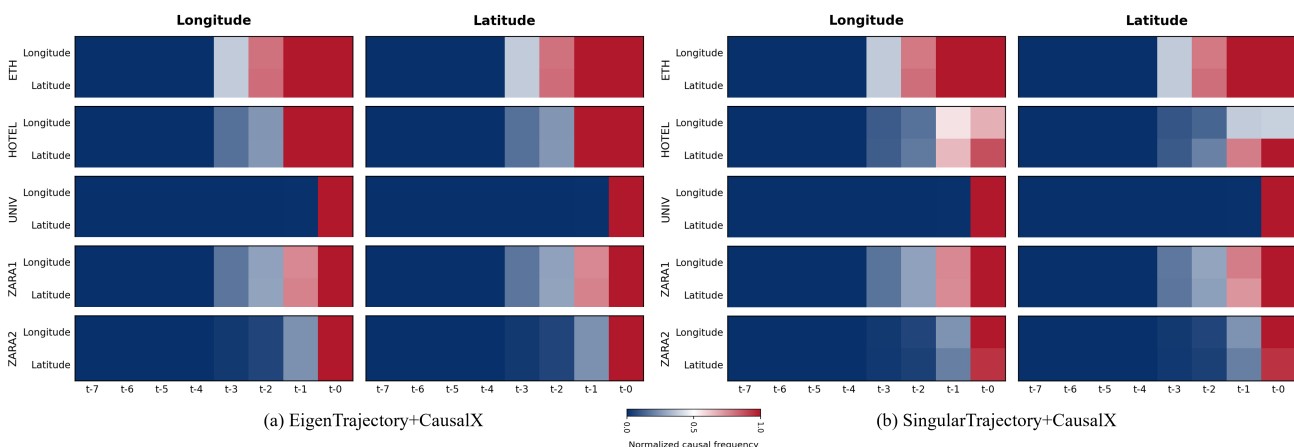

*Figure 10.* Another type of causal graphs showing the normalized causal frequency from all historical nodes to each target variable for pedestrian prediction. Rows correspond to the two target variables (Longitude, Latitude), and columns represent source timesteps from $t-7$ to $t-0$.

**Backbone selection rationale.** To provide a broader evaluation beyond a single architecture or task, we report results on *two downstream forecasting domains* (tropical cyclones and pedestrians). Within pedestrian forecasting, we choose EigenTrajectory and SingularTrajectory to cover the two representative integration types supported by CausalX: *graph-based* and *feature-based* incorporation of causal information, where EigenTrajectory corresponds to the graph-form instantiation. This selection aims to test the generality of the graph-editing faithfulness trend across both task settings and architectural variants.

## C. Additional Visualization

This section presents additional visualizations illustrating the interpretability of CausalX.

**Additional visualization of learned causal-inspired graphs.** As mentioned in Fig. 4 of the main paper, we provide here the full causal-inspired graphs covering all historical timesteps and all modalities. Fig. 10 and Fig. 11 visualize the normalized causal frequencies from every source node to each target variable. These complete visualizations offer a more comprehensive picture of how CausalX integrates temporal and cross-variable causal cues.

For **pedestrian forecasting** (Fig. 10), both EigenTrajectory+CausalX and SingularTrajectory+CausalX exhibit highly consistent and interpretable causal patterns across datasets. As expected, longitude and latitude at $t-0$ dominate the causal influence, while contributions decay smoothly with increasing temporal distance, matching the short-horizon dynamics of human motion. CausalX further captures *dataset-specific* structures: HOTEL and ZARA1 display more mid-range dependencies (around $t-3$ to $t-2$), reflecting their more complex crowd flows. Importantly, these patterns emerge *automatically*, demonstrating that CausalX reliably learns motion principles that are both human-interpretable and consistent across backbones.

For **tropical cyclone forecasting** (Fig. 11), CausalX reveals substantially richer and physically aligned causal structures. TC-Diffuser+CausalX identifies intricate cross-variable interactions: future track (longitude/latitude) depends not only on recent positions but also on pressure, wind speed, geopotential height, and historical intensity changes. Meanwhile, $\text{TCN}_M$+CausalX shows more coherent long-range dependencies, especially for pressure and wind speed, closely matching known geophysical processes in cyclone evolution. What is notable is that both backbones, despite their architectural differences, converge to similar high-level causal patterns under CausalX guidance. This highlights its ability to impose a *stable, domain-consistent, and physically meaningful causal prior* across model families.

Taken together, these full-graph visualizations demonstrate that CausalX does not merely improve predictive accuracy: it learns domain-consistent dependency structures that are prediction-critical and human-interpretable. The learned causal structures are (1) *interpretable*, (2) *domain-aligned* (matching real-world motion and geophysical behaviors), (3) *consistent across backbones*, and (4) *robust across datasets*. Such qualities underline the generality and reliability of CausalX as a plug-and-play causal enhancement model for sequential forecasting tasks.

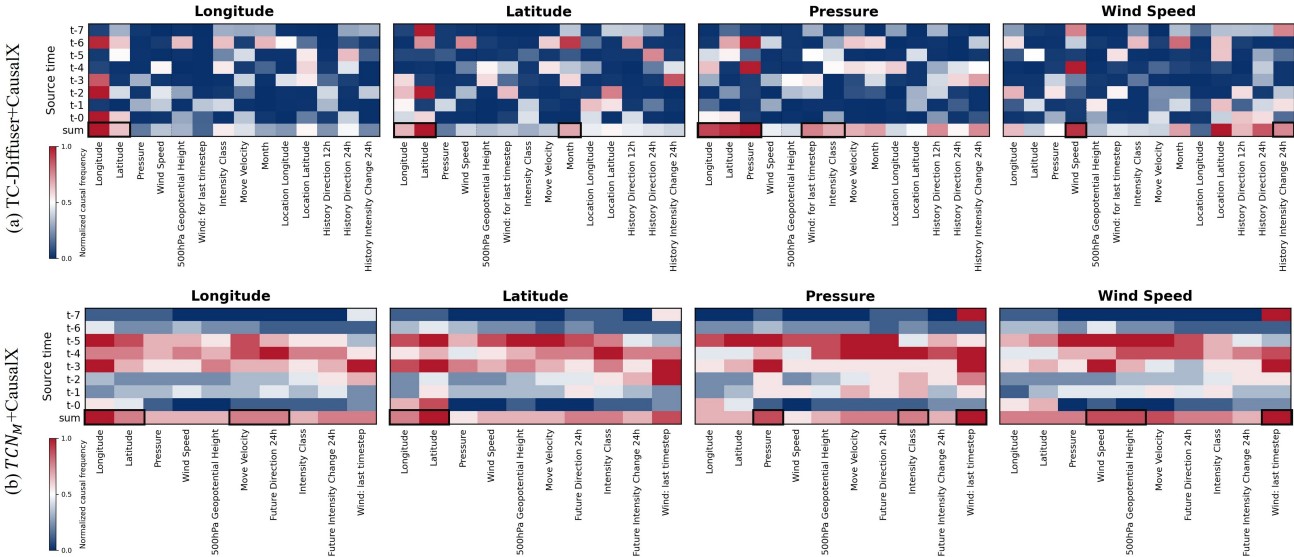

*Figure 11.* Another type of causal graphs illustrating the normalized causal frequency from all historical nodes to each target variable for tropical cyclone forecasting. For each predicted variable (Longitude, Latitude, Pressure, Wind Speed), rows correspond to historical timesteps $t-7$ to $t-0$, while columns indicate the contributing source features (e.g., position, pressure, wind speed, geopotential height, motion indicators, historical intensity change). The bottom row ("sum") shows the average causal contribution aggregated over all eight historical timesteps, reflecting the overall importance / salience of each meteorological modality.

**Additional visualization of multi-source causal graphs.** To elucidate what each constraint emphasizes and how they complement one another, Fig. 12 shows chord diagrams for the four supervision graphs and the final causal-inspired graph.

**(a)** *Granger.* As a predictive test, it concentrates mass on *near* lags ($t - 1/t - 2$); e.g., `Pressure_5/6` and `WindSpd_5/6/7` strongly point to the targets, reflecting that very recent history provides the strongest evidence for forecasting.

**(b)** *Do-calculus.* Feature-level interventions yield more conservative magnitudes, yet follow the same *short-lag* pattern as *Granger* (e.g., `Longitude_5`, `Pressure_6`), indicating that the encoder captures the expected temporal locality of influences.

**(c)** *TDMI.* Time-Delayed MI highlights *lagged* dependencies; salient edges arise from mid-range delays ($t - 3 \sim t - 5$), consistent with delayed responses in TC evolution (e.g., `Longitude_3/4`, `WindPrev_4/5`).

**(d)** *VAE.* The generative constraint produces the *most diverse* cross-node patterns (e.g., `month_5`, `MoveVel_1{5`, `z500_6`), capturing multi-modal couplings beyond immediate temporal adjacency.

**(e) Final graph.** Combining these complementary cues yields the most complete structure: recent position/motion dominate longitude/latitude; intensity/pressure history drives pressure; and wind history governs wind speed. The fused graph recovers physically plausible drivers while offering finer connectivity than any single constraint.

In summary, across the four constraints, *Granger* and *do-calculus* emphasize short-lag predictive effects, *TDMI* surfaces mid-lag temporal dependencies, and the *VAE* captures diverse cross-modal couplings; fusing them yields a physically consistent, fine-grained causal-inspired graph that is more complete and interpretable than any single source, underscoring that each module contributes a distinct, necessary signal.

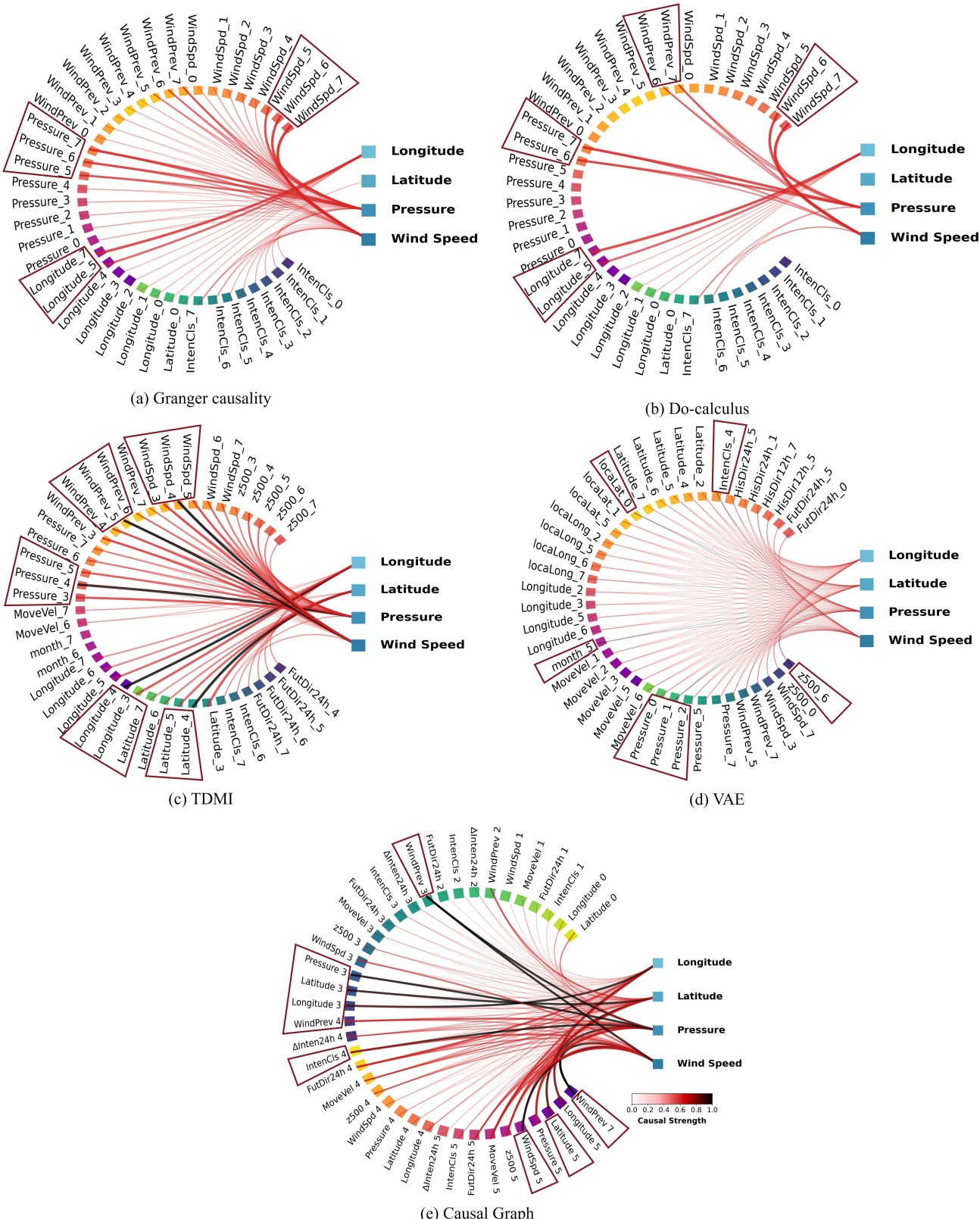

*Figure 12.* **Causal chord diagrams for each supervision causal loss**. **(a)** *Granger*, **(b)** *do-calculus*, **(c)** *TDMI*, **(d)** *VAE*, and **(e)** the final fused graph on TC-Diffuser. Each constraint stresses different aspects (predictive, interventional, temporal, generative), and their fusion yields a more complete and physically consistent causal structure.

