# OpenReview forum: "CausalX: A Unified and Causally-Interpretable Plug-and-Play Model for Multi-modal Spatio-Temporal Forecasting"
_ICML.cc/2026/Conference — ICML 2026 regular_

### Official Review · Reviewer_zNSc · 2026-03-04

**Soundness:** 2
**Presentation:** 2
**Significance:** 2
**Originality:** 2
**Overall Recommendation:** 4
**Confidence:** 4

**Summary:**

This paper studies the multi-modal spatio-temporal forecasting problems. The authors claim that current methods have an interpretability problem. To address this gap, they introduce CausalX, a causally interpretable plug-and-play approach for multi-modal spatio-temporal forecasting. Through creating a dynamic causal graph and a VAE, CausalX achieves interpretability without the ground truth causal graph. Experiments on the real-world spatio-temporal forecasting tasks confirm the performance and interpretability of CausalX.

**Compliance With Llm Reviewing Policy:**

Affirmed.

**Final Justification:**

The authors' rebuttal has addressed my concerns. Therefore, my final assessment is weak accept (4).

**Key Questions For Authors:**

Do the authors originally define the Spatio-temporal time series forecasting problems? If not, please cite properly.

**Limitations:**

There is a lack of introduction to the critical concepts and related citations. I suggest that authors add properly.

**Strengths And Weaknesses:**

Strengths:
1. The studied spatio-temporal forecasting problem is practical and essential in real-world applications.

2. The overall logic of the introduction section is sound.

3. The experiments are comprehensive, authors compare with multiple existing methods.

Weaknesses:
1. There is no related work section in the main text. I understand this may be because of the length constraint of the main text. However, as an important and necessary part of the research paper, I suggest that authors write a related work section and put it in the main text or the appendix.

2. Since the spatial-temporal forecasting is an active and important area, and this work is inspired by causality. I would encourage authors to discuss more works of spatial-temporal causal inference, which include (but are not limited to): [1] and [2].

3. I think there may be some problems with the preliminary (or problem formulation, I think). Specifically, as the problem is spatial-temporal series prediction, why is it a 1D-to-1D problem? Typically, I believe a spatial-temporal series may be a high-dimensional series. For example, the series of a spatial distribution.

4. I suggest that authors add some related critical causal concepts in the preliminary section. For example, the structural causal models (SCMs).

5. Some citations of the important concepts or methods are missing. For example, the Granger causality has no citations.

6. There is no identifiability proof of the causal structures. I suggest the author discuss the identifiability of the causal graph under a spatial-temporal setting.

[1] Li H, Chi H, Liu M, et al. Transformer-Based Spatial-Temporal Counterfactual Outcomes Estimation[C]//Forty-second International Conference on Machine Learning.

[2] Christiansen, R., Baumann, M., Kuemmerle, T., Mahecha, M. D., and Peters, J. Toward causal inference for spatio-temporal data: conflict and forest loss in colombia. Journal of the American Statistical Association, 117(538): 591–601, 2022.

---

> ### Author Rebuttal · Authors · 2026-03-29
>
> Thank you for recognizing the **practical importance, sound overall logic, and comprehensive experiments** in our work, and for your valuable comments and suggestions. These comments help us improve the clarity and positioning of the manuscript. We will incorporate the corresponding revisions in the future version. Below are our responses.
>
> **Q1: Do the authors originally define the Spatio-temporal time series forecasting problems? If not, please cite properly. I think there may be some problems with the preliminary. Specifically, as the problem is spatial-temporal series prediction, why is it a 1D-to-1D problem? Typically, I believe a spatial-temporal series may be a high-dimensional series. For example, the series of a spatial distribution.**
>
> **A1:** No, we do **not** originally define it, and we will cite it properly. As noted in the reviewer-suggested literature, a spatio-temporal dataset can be viewed as a realization of a spatio-temporal stochastic process observed at discrete points in space-time. We will add this citation and revise the preliminary accordingly. To clarify the “1D-to-1D” description:
> - (1) This mainly refers to the **core sequential target/variable** (e.g., historical trajectory), rather than implying that the entire model only takes 1D input. In practice, it also uses **multi-modal inputs**, including 2D visual/spatial observations, rather than only 1D signal.
> - (2) More importantly, **the task remains spatio-temporal because the variables evolve over time and are associated with spatial locations**, movement, interactions, or context. Thus, even when the final prediction target is low-dimensional, the problem is not equivalent to ordinary 1D forecasting such as stock prediction. For example, pedestrian or TC trajectory forecasting involves not only temporal evolution but also spatial movement.
>
> We will revise the text to clarify this definition and avoid the misleading “1D-to-1D” impression.
>
> **Q2: There is no related work section in the main text. I understand this may be because of the length constraint of the main text. I suggest that authors write a related work section.**
>
> **A2:** Thank you for your understanding. We will add a clearer related-work discussion to better position our method with respect to multi-modal spatio-temporal forecasting, interpretability, and spatio-temporal causal inference.
>
> **Q3: Since the spatial-temporal forecasting is an active and important area, and this work is inspired by causality. I would encourage authors to discuss more works of spatial-temporal causal inference, which include (but are not limited to): [1] and [2].**
>
> **A3:** We agree and will add this discussion. Christiansen et al. provide a formal causal framework for spatio-temporal processes, while noting that the full causal structure is often difficult to specify. Li et al. formulate a spatio-temporal counterfactual estimation problem and study identifiability of the estimation targets under assumptions, rather than recovery of a unique ground-truth causal graph. This is aligned with our view that , the goal is often not to recover a fully specified true graph in realistic spatio-temporal settings, but to learn a structured representation that is causally meaningful and practically useful. We will add this discussion in the revision to better position our method.
>
> **Q4: Add some related critical causal concepts.**
>
> **A4:** Thanks for your suggestion. We will add a brief introduction of key causal concepts such as SCMs in the preliminary section.
>
> **Q5: Citations.**
>
> **A5:** Thank you for pointing this out. We will carefully check the manuscript and add the missing foundational citations, including Granger causality and other under-cited methods.
>
> **Q6: There is no identifiability proof of the causal structures. I suggest the author discuss the identifiability of the causal graph under a spatial-temporal setting.**
>
> **A6:** We thank the reviewer for raising this important point. We agree that the current paper does not provide an identifiability proof for the causal structures, as **strict identifiability of the true causal graph is highly challenging and difficult to guarantee in such complex multi-modal spatio-temporal forecasting settings**. This is also consistent with the reviewer-suggested literature: Christiansen et al. explicitly note that the full causal structure of a spatio-temporal process is difficult to fully specify, while Li et al. focus on identifiability of estimation targets under assumptions rather than unique graph recovery.
>
> Therefore, our goal is not to recover the one true spatio-temporal causal graph. Rather, we use multiple causal-inspired proxies to learn a **more structured, interpretable, and forecasting-helpful dynamic graph**. We will revise the manuscript to make this limitation explicit.
>
> *If the reviewer has further questions or suggestions, we would be very happy to discuss them during the author-reviewer discussion period.*

---

> > ### Author Rebuttal · Reviewer_zNSc · 2026-04-01
> >
> > Thanks to the authors' detailed response, most of my concerns have been addressed. Especially the author’s discussion of the spatial-temporal causal inference. I also believe that adding a Discussion can better position your method. I have increased my score to a weak accept (4).

---

> > > ### Author Response · Authors · 2026-04-01
> > >
> > > Thank you very much for your time, careful consideration, and encouraging update. We are very glad to know that our rebuttal addressed most of your concerns, especially regarding the discussion of spatial-temporal causal inference and the positioning of our method. Your comments have been very valuable in helping us strengthen the paper. We truly appreciate your constructive feedback and support, and we will incorporate the corresponding improvements into the revised version.

---

### Official Review · Reviewer_y5Jq · 2026-03-12

**Soundness:** 2
**Presentation:** 1
**Significance:** 1
**Originality:** 3
**Overall Recommendation:** 4
**Confidence:** 4

**Summary:**

This paper describes a framework for causal inference for spatio-temporal signals. CausalX produces a time-varying graph of the causal relationships among the variables both in the time dimensions and the dimension of the observables. The graph is produced in three stages. First, initial node features are fed into a Graph Attention Network that produces an updated node embedding and estimated causal strength estimates on the edges. Then  various causal reasoning techniques (Granger, do-calculus, TDMI, VAE) are employed that each produce an estimated causal graph, and enable a refinement of the causal graph. Finally, a diffusion process is used to finalize the graph. The output of this pipeline - node and graph level features, as well as the causal graph itself, may be fed into other modules for downstream tasks such as prediction, classification, etc.

**Compliance With Llm Reviewing Policy:**

Affirmed.

**Final Justification:**

They address most of my concerns, and I am happy to raise my score to 4.

**Key Questions For Authors:**

See the detailed comments in the Strengths And Weaknesses.

**Limitations:**

yes

**Strengths And Weaknesses:**

Strengths: It is interesting that CausalX can exploit existing causal analysis approaches including do-calculus, TDMI, and Granger causality, and the integration of those approaches with recent generative modeling techniques such as diffusion models, as well as architectures like graph neural networks. The experimental evaluation is comprehensive and the appendix gives a significant level of detail for potential reproducibility.

Weaknesses: While this work describes a pipeline that appears to yield promising results on several benchmark problems, it is difficult to glean any new insight into causal analysis from it. Certain experimental results hint at potential explanations, but these are not analyzed. For instance, in the ablation study, it is suggested that Granger causality is the most important contributor in one setting, while the do-calculus is important in another. A potentially fruitful research question may involves attempting to understand from a theoretical perspective why that is so.

Throughout, the discussion of related work, including citations, has several issues. In general, there is a lack of citations of the various technical tools involved. For instance, in Section 3, around line 89, various terms including Granger causality,  time delayed mutual information, do-calculus, and variational autoencoders, are used without definition or citations. Later, Section 3.2 leverages "conditional diffusion-based graph denoising process" and DDPM and citations or definitions should be given for these as well.  Additionally, there are multiple papers in the bibliography that are not cited in the text body, one of which is seemingly unrelated to the manuscript, including "Crafting papers on machine learning." by Langley, P.

In Section 3.1, the definitions of the various vectors and formulas could benefit from clarification:
- The dimensionality for the various tensors frequently includes B but this variable is not defined.
- Additionally, the graph attention network produces Z, edge, and \alpha. What are the dimensionalities of edge and \alpha? It’s stated (on line 156) that CG_0 = (edge,  \alpha) and later near 167 that CG_0 \in Bx(TN)^2, and difficult to reconcile these two formulas.

Minor comments:
- Section 3.1.3: Mutual information is defined for variables with known densities or probability mass functions. The authors may want to clarify how they estimate MI in section 3.1.3. There appears to be a relevant paper in the bibliography ("Estimating mutual information." by Kraskov et. al), but it is not cited in the text body.
- Section 3.1.2.  It is stated that interventions are simulated by "mean scaling", without providing a definition of citation.
- The title of this paper may lead to confusion. The technical content of this work is largely in the area of causal analysis, and less so in the area of forecasting. The benefit for forecasting is that CausalX would provide more useful signals for existing forecasting models. Furthermore, "plug-and-play" suggests to the reviewer that CausalX integrates with existing forecasting pipelines without retraining. However, this does not appear to be the case, as the methodology described involves retraining the downstream predictors (line 691). This interpretation of "plug-and-play" should be clarified early in the paper.

---

> ### Author Rebuttal · Authors · 2026-03-29
>
> Thank you for recognizing the **integration of multiple causal analysis approaches, the comprehensive experiments, and the reproducibility details** in our work. We also appreciate your suggestions on positioning, presentation, notation, and citations. Below are our responses.
>
> **Q1: While the results are promising, it is difficult to glean any new insight into causal analysis. The ablations suggest different causal components matter in different settings, but this is not analyzed.**
>
> **A1:** Thank you for this insightful suggestion. We agree that why different causal proxies play different roles across scenarios is an important question for further study. What we would like to clarify is that our current focus is **not** to propose a new standalone causal analysis theory, but to develop a unified framework that integrates multiple **causal-inspired** signals into dynamic graph learning for complex multi-modal spatio-temporal forecasting. This is motivated both by the practical need in such high-risk settings and by the fact that a fully specified ground-truth causal graph is generally unavailable and difficult to verify (also supported by related literature; see our response to Reviewer zNSc, **A3**). Therefore, our goal is not to recover one true graph, but to learn a more structured, interpretable, and forecasting-helpful dynamic graph.
>
> That said, the current ablations already suggest a plausible explanation: different proxies impose different **relational inductive biases**. Granger emphasizes predictive dependencies, TDMI delayed temporal dependence, do-calculus node-to-node effects, and VAE generative consistency. Therefore, different scenarios may favor different proxies depending on their dominant dependency patterns. We will clarify this interpretation and position it as a future direction for deeper theoretical analysis.
>
> **Q2: Related-work discussion and citations. Some concepts are introduced without citation. Uncited bibliography items.**
>
> **A2:** We will revise this carefully. We will add missing foundational citations and brief definitions when these concepts first appear, including **Granger causality** (Granger, 1969), **do-calculus** (Pearl, *Causality*), **VAE** (Kingma and Welling, 2014), **DDPM** (Ho et al., 2020), and expand the related-work discussion. The unrelated bibliography item came from an ICML template entry, and we will remove all uncited or irrelevant references. Thank you for pointing this out.
>
> **Q3: Section 3.1: B is undefined, and the shapes of $Z$, `edge`, `α`, and $CG_0$ are hard to reconcile.**
>
> **A3:** Sorry for confusion. **B denotes the batch size**. The current text refers to two valid representations of the same graph structure. Specifically, $CG_0$ = (`edge`, `α`) refers to the **sparse graph representation**, where `edge` stores the edge index pairs and `α` the corresponding edge strengths. By contrast, $CG_0$ ∈ $R^{B\times(TN)\times(TN)}$ refers to the **corresponding matrix form** after mapping these edge strengths back to the full node-pair space. For example, with \(T=1\) and \(N=2\), there is an edge from node 1 to node 2, this edge is stored in `edge`, and its corresponding strength is stored in `α`; this can then be converted into a $2\times2$ matrix form. We will revise this section.
>
> **Q4: MI is used, but its practical estimation procedure is not explained.**
>
> **A4:** In our implementation, TDMI is computed using `mutual_info_score` from the Python package `sklearn` after discretization, and we will state this more clearly.
>
> **Q5: “Mean scaling” is used to simulate interventions, but it is not defined.**
>
> **A5:** Thank you for pointing this out. In our implementation, “mean scaling” simulates an intervention on node \(i\) by replacing its feature vector with the sample-wise mean feature vector across nodes, multiplied by a perturbation scale. We then re-run the graph model and measure the average absolute change in downstream node embeddings relative to the original output. This change is used as a proxy for the interventional influence of node \(i\) on other nodes. We will make this definition explicit and clarify that it is an intervention-inspired perturbation mechanism for graph learning.
>
> **Q6: Title: “Plug-and-play” suggests integration without retraining, but the method appears to require retraining of downstream predictors.**
>
> **A6:** We understand that **“plug-and-play”** can be misunderstood as **zero-retraining insertion**, which is not our intended meaning. What we mean is **modular adaptation across backbones** through generic mechanisms such as feature or graph augmentation. When the target backbone has a different input organization, some retraining is naturally required; when similar tasks share aligned inputs, adaptation is lighter (see reviewer b6KU **A2** for details). Therefore, “plug-and-play” in our paper should be understood as **modular and lightweight adaptation**.

---

> > ### Author Rebuttal · Reviewer_y5Jq · 2026-04-03
> >
> > Thank you for the clarification on notations. If I understand it correctly, the authors present $(\alpha)$ as a causal graph, yet Eqs. (6), (8), (10), and (12) regress (\alpha) toward four heterogeneous proxies but without defining a common latent estimand. I have a concern about Eq. (5), because it uses $(1-\min_\ell p_\ell)$, the expected score under the null increases with the number of tested lags, so non-causal relations can receive large “causal” scores. In addition, Granger and TDMI collapse lag information before broadcasting to the full time-expanded graph, the do-calculus term is a perturbation sensitivity measure rather than an identified intervention effect, and the VAE score cannot distinguish direct causation from common-cause correlation. The diffusion module is trained only to denoise $(CG_0)$, I am unclear why it would recover missing causal structure rather than a smoothed version of the initial graph.
> >
> > One minor thing, my understanding is that one must add $(L_{\text{CausalX}})$ to the backbone loss and train the integrated system end-to-end from scratch. It also requires offline Granger/TDMI precomputation and a hand-crafted prior graph $(G_{\text{prior}})$. That is modular training-time augmentation, not plug-in insertion in the usual sense. Am I missing something here?

---

> > > ### Author Response · Authors · 2026-04-04
> > >
> > > Thank you for the thoughtful follow-up. We are glad that our clarifications on the other points did not raise further concerns. Regarding the remaining issue, we would like to make our position explicit:
> > > - First, **our goal is not to recover the true causal graph.** Rather, our goal is to learn a more structured, interpretable, and useful graph by combining **multiple complementary views**. The added evidence on asymmetry, stability, and faithfulness (see Reviewer aJix A1/A3/A4), together with the visualizations (Figs. 5–6), the Top-k / Random-k / Keep-only-Top-k analyses (see Reviewer aJix A4), and the forecasting performance results, all support that the learned graph achieves this goal.
> > > - **`α`**: The heterogeneous proxies in Eqs. (6), (8), (10), and (12) **jointly constrain a shared learnable edge score**  `α`, which is the unified graph parameter in our model. In this sense,  `α` is a shared relational score shaped by multiple proxies. For example, $CG_{\text{Granger}}$ is a fixed proxy graph, while the optimization target remains the learned  `α`; the corresponding loss term regularizes  `α` toward that proxy.
> > > - **Eq. (5)**: The lag range is fixed and minimal, with **max\_lag = 1**, so the concern about inflation from searching over many candidate lags is substantially reduced here. Correspondingly, the Granger/TDMI term should be understood as providing a **coarse temporal structural bias**.
> > > - The **do-calculus term** is not intended as an identified intervention effect, but rather as a **semantically aligned, node-level perturbation-based relational proxy** in the same GAT semantic space (see Reviewer aJix **A2**): it perturbs a node-aligned representation, measures how that perturbation propagates to other nodes, and uses the resulting pairwise effect matrix to regularize the learned graph.
> > > - **Diffusion module**: It provides a **refinement space** in which the proxy-guided initial graph can be further adapted under the downstream task objective, enabling **task-aware graph refinement**.
> > >
> > > Accordingly, we do **not** intend to claim that regressing a learned graph provides a formal guarantee of recovering missing true causal structure. Rather, each proxy captures only part of the desired structure, and our purpose is to **combine these partial structural biases** into a more directional, stable, interpretable, and practically useful graph. In this sense, the learned graph should not be understood as **merely a smoothed version of the initial graph**: unlike simple smoothing, which mainly preserves and regularizes the original graph, our graph is learned under heterogeneous proxy constraints and further refined by the downstream task objective. This interpretation is also supported by our **empirical results**:
> > > - Reviewer aJix **A1** compares the initial graph $CG_0$ and the final graph, showing that the latter is not merely a smoothed version of the former: it is substantially more **directional**, with much higher asymmetry and a much larger proportion of clearly one-directional edges, consistent with the temporal ordering of earlier-to-later influence.
> > > - The full ablation in Table 5 shows that each proxy provides a **non-redundant contribution**, since removing any one of them degrades performance.
> > > - The learned graph exhibits strong structural properties, including **stability** and **faithfulness** (see Reviewer aJix **A3/A4**).
> > >
> > > Overall, we will revise the manuscript to make it clearer that CausalX should be understood as a **causally informed, proxy-guided, structured graph-learning framework**, rather than a formally identifiable causal discovery method or a simple graph-smoothing mechanism.
> > >
> > > On the “plug-and-play” point, we will revise the wording explicitly. In our paper, **“plug-and-play” means module-level transferability across backbones**, rather than direct full-parameter transfer or zero-cost insertion into an already trained pipeline. In the most general case, your understanding is correct: the current framework is closer to **modular training-time augmentation** than plug-in insertion without retraining in the usual sense.
> > >
> > > At the same time, we would still like to reiterate that, **under the setting described in our first-round rebuttal, the adaptation burden can remain relatively small when the data/input configuration is the same**. Concretely, the proxy-graph generation process remains unchanged, so the offline Granger/TDMI computations and prior graph can be reused directly without recomputation; in such cases, one may reuse the learned front part (including the proxy-graph generation pipeline) and directly insert the diffusion-refined graph into the original backbone, with retraining reduced to optional or lightweight adaptation, depending on the downstream integration design. We will further refine the word in the revised version to better position plug-and-play.
> > >
> > > Thank you again for this helpful follow-up; we hope these clarifications help reduce your concern.

---

### Official Review · Reviewer_aJix · 2026-03-18

**Soundness:** 2
**Presentation:** 3
**Significance:** 3
**Originality:** 2
**Overall Recommendation:** 2
**Confidence:** 3

**Summary:**

This paper introduces CausalX, a plug-and-play module for multi-modal spatio-temporal forecasting that learns instance-specific dynamic causal graphs over modalities and time. The proposed method regularizes a GNN-derived attention graph with multiple causal signals—Granger causality, TDMI, do-calculus-inspired interventions, and a VAE-based generative constraint—and then further refines the graph with a prior-guided diffusion process. The learned graph or graph representation is fused for downstream forecasting. Experiments are reported on pedestrian trajectory prediction and tropical cyclone forecasting.

**Compliance With Llm Reviewing Policy:**

Affirmed.

**Final Justification:**

I would like to maintain my score mainly because I did not see an elegant design and justification (or proof) of why regressing one graph over several heterogeneous proxy should guarantee the learning of some meaningful causal structures in complex spatio-temporal setup (where there may be a lot of instaneous causalities, confounding factors, complex latent dynamics).

**Key Questions For Authors:**

1. What exactly should readers interpret the learned edge weights as: causal effects, causal attributions, or predictive dependencies? The current wording seems stronger than the evidence.
2. How is the do-calculus component implemented, and in what sense does it approximate intervention semantics rather than feature ablation?
3. How stable are the learned graphs across random seeds and different training splits?
4. Can the authors provide a more quantitative explanation-faithfulness test beyond visual inspection and performance drop after edge removal?

**Limitations:**

see above

**Strengths And Weaknesses:**

Strengths
+ Interpretability in high-stakes forecasting is valuable, and the paper targets a real gap between strong black-box forecasters and methods that can explain cross-modal and temporal dependencies in a structured way.
+ It combines several weak causal signals rather than relying on a single proxy.

Weakness
- The learned graph is built from attention weights and regularized by several heuristic or proxy signals, but this does not establish that the resulting edges correspond to identifiable causal effects in the usual sense. In particular, combining Granger, TDMI, intervention-style masking, and VAE reconstruction may yield a useful dependency graph, but they do not convincingly justify why this should be interpreted as a causal graph rather than a structured attribution graph.
- Evaluation shows gains, but most of them are incremental rather than dramatic, and the interpretability claims are still somewhat qualitative.
- The “plug-and-play with little overhead” claim is somewhat overstated. The efficiency table shows noticeable training-time increases for several backbones, including 5h to 7h, 13h to 16h, 20h to 28h, and even more than doubling from 20min to 45min for one pedestrian model. Inference overhead is indeed small, but the training overhead is not negligible.

---

> ### Author Rebuttal · Authors · 2026-03-29
>
> Thank you for recognizing the **importance of interpretability**, the **practical value of the problem**, and the benefit of **combining multiple weak causal signals**. We appreciate your comments on interpretation, faithfulness, and efficiency. Below are our responses.
>
> **Q1: How should the learned edge weights be interpreted?**
>
> **A1:** The learned edge weights are best viewed as a **causally-inspired relational structure**: stronger than predictive dependencies or attention weights, but not strictly identifiable causal effects. Our goal is to integrate multiple causal-inspired proxies into dynamic graph learning where no ground-truth causal graph is available (see details in Reviewer y5Jq **A1**), making the learned graph more **directional, structured, interpretable, and useful for forecasting**. It shows **causal-graph-like properties**:
>
> - **Directional structure.** Compared with the initial attention graph $CG_0$ on the full TCN$_M$ test set, the final learned graph has much higher **asymmetry** ($0.69 \pm 0.07$ vs. $0.34 \pm 0.24$; larger means less symmetric and thus more directional), and a much higher proportion of edges that are clearly **one-directional rather than roughly balanced in both directions** ($0.82 \pm 0.09$ vs. $0.35 \pm 0.27$). This is reasonable in a temporal graph, where earlier-to-later influence is naturally stronger than reverse-direction influence.
>
> - Our added results in **A3** and **A4** further support **stability** and **faithfulness**.
>
> Overall, we will revise the wording to a **causal-inspired graph**.
>
> **Q2: How is the do-calculus implemented, and why approximate intervention semantics rather than feature ablation?**
>
> **A2:** For each node $i$, we apply a controlled perturbation (mean scaling; see Reviewer y5Jq **A5**) to its representation $F_i$, keep all other components unchanged, rerun the same GAT to obtain $Z^{per(i)}$, and measure the downstream change $\lvert Z_j^{per(i)} - Z_j \rvert$ as the effect score $CG_{do}(i \to j)$. This pairwise effect matrix then regularizes the learned attention $\alpha_{(i,j)}$, so this component approximates intervention semantics through a **node-level perturbation mechanism**.
>
> Why this is more than feature ablation:
>
> - it perturbs the **node-aligned representation** $F_i$, not arbitrary hidden units, so it has clear **node-level semantics**;
> - it measures how the perturbation **propagates to other nodes**, yielding a **node-specific pairwise relational effect**, not just a global change;
> - $CG_{do}$ and the learned graph are defined in the **same GAT semantic space**, so this is an internally aligned relational proxy rather than an external ablation signal.
>
> **Q3: Graph stability**
>
> **A3:** The learned graphs are **stable**. We use **Spearman rank correlation** for full edge-ranking similarity and **Top-30% Jaccard overlap** for salient-edge overlap. We report results on SingularTrajectory. Using 8 learned graphs (5 seeds, 3 training splits), we compute all 28 pairwise comparisons. The mean pairwise Spearman rank correlation is **0.87 ± 0.06**, and the Top-30% Jaccard overlap is **0.74 ± 0.07** (both with **1.0** as the maximum; higher is better), showing that both the overall edge ranking and salient-edge set are largely preserved.
>
> **Q4: More quantitative explanation-faithfulness test**
>
> **A4:** Beyond the edge-removal results in our paper, we add a **sufficiency** test via **keep-only top-k** graph editing. **Please see the supplementary figure:** `https://anonymous.4open.science/r/CausalX-1B37/README.md`. **Top-k keep** consistently yields the **smallest** performance drop among all settings (Top-k remove / Random remove / Top-k keep). On TCN$_M$, Top-k keep gives the smallest relative drop, only **2.36** for Track, versus **14.26** for Top-k remove and **7.84** for Random remove, providing additional evidence of **strong explanation faithfulness**.
>
> **Q5: Most of the improvements are incremental rather than dramatic.**
>
> **A5:** The gains on some pedestrian prediction backbones are moderate, but they should be interpreted in the context of already strong baselines, where **published methods often differ only by small margins**; thus, even modest but **consistent improvements** remain meaningful. Beyond accuracy, CausalX provides a causal-inspired graph, improving interpretability.
>
> **Q6: "Plug-and-play with little overhead” claim is somewhat overstated for training.**
>
> **A6:** In Fig. 8, although CausalX increases the per-epoch cost, it reaches the **backbone’s best baseline level earlier** (e.g. SingularTrajectory: **10 min vs. 15.5 min**). Reaching its own peak can be slightly later for some backbones (e.g., SingularTrajectory+CausalX: **16.5 min vs. 15.5 min**), which we view as a reasonable training-time trade-off for improved final accuracy. TC-Diffuser is an exception: +CausalX reaches its own higher peak earlier (**7 h vs. 7.25 h**).
>
> *We would be happy to discuss further questions during the discussion period.*

---

> > ### Author Rebuttal · Reviewer_aJix · 2026-04-03
> >
> > Thank you for your efforts to clarify. The rebuttal clarifies that the learned graph is only “causal-inspired,” which reinforces my main concern: the paper does not provide a theoretical justification for why regressing one graph against several heterogeneous proxy should recover missing causal structure? The added evidence on asymmetry, stability, and faithfulness only shows that the learned graph is directional and useful for prediction, not that the fusion of Granger, TDMI, perturbation effects, and VAE scores has any guarantee of identifying a meaningful causal graph. As a result, I still do not find the central causal claim convincing and will maintain my score.

---

> > > ### Author Response · Authors · 2026-04-04
> > >
> > > Thank you for your reply and for further clarifying your remaining concern. We are also glad that our clarifications on the other points did not raise further concerns. Regarding your current concern — namely, **why regressing a learned graph against several heterogeneous proxies should recover missing causal structure** — we would like to make our position explicit:
> > >
> > > **Our claim is not that regularizing a learned graph with multiple heterogeneous proxies can formally recover the true missing causal structure.** More fundamentally, our goal is **not** strict causal graph identification. Rather, our goal is twofold: (1) to provide a **more organized and interpretable account** of cross-modal and temporal dependencies in a highly complex multi-modal spatio-temporal forecasting setting, and (2) to help the model capture forecasting-helpful relational structure that **improves prediction performance**.
> > > Under this goal, we believe the current method already achieves these two practical objectives to a substantial extent. Specifically, our method uses multiple complementary proxies to guide graph learning toward a graph with useful causal-inspired properties. In particular, the added evidence on **asymmetry, stability, and faithfulness**, together with the visualization results and the **Top-k / Random-k / Keep-only-Top-k** analyses, supports that the learned graph provides a more structured and interpretable account of cross-modal and temporal dependencies, while also being more useful for prediction.
> > >
> > > Besides, we hope to clarify the role of multiple proxies as follows:
> > >
> > > - **Our goal is to combine partial structural biases, not to prove causal recovery.** In complex multi-modal spatio-temporal forecasting, any single proxy captures only part of the desired structure. Our motivation for combining Granger, TDMI, perturbation effects, and the generative constraint is therefore to aggregate these partial structural biases, so as to learn a more structured, stable, and useful structured relational graph under severe non-identifiability, rather than to claim recovery of the true causal graph.
> > >
> > > - **Fig. 12 illustrates that different proxies contribute different views of the desired relational structure.** The individual proxies emphasize different aspects of the desired structured relational graph (e.g., predictive, intervention-inspired, temporal/nonlinear, and generative consistency), and each alone yields a relatively one-sided structure. In this sense, the purpose of fusion is **not** strict causal recovery, but to combine complementary views so as to regularize graph learning toward a more directional, structured, interpretable, and useful relational graph.
> > >
> > > - **The proxies are empirically complementary and necessary rather than interchangeable.** This is also supported by the full ablation in **Table 5**: removing any one proxy degrades performance, showing that each proxy contributes non-redundant gains and that the design does not rely on any single interchangeable heuristic.
> > >
> > > Thus, our claim is not that heterogeneous proxy fusion provides a strict theoretical guarantee for recovering missing causal structure. Rather, given that a single proxy is insufficient to characterize such a complex multi-modal forecasting setting, that no ground-truth causal graph is available, and that each proxy contributes measurable performance gains, we believe the multi-proxy design is both necessary and effective as a practical way to learn a more structured and useful graph.
> > >
> > > We fully agree that this scope boundary should be stated more explicitly. We will further revise the manuscript to avoid any implication of guaranteed causal recovery, and to present the method more precisely as a **causal-inspired structured graph-learning module** rather than a method with formal causal identification guarantees. Thank you again for helping us clarify this important boundary.

---

### Official Review · Reviewer_b6KU · 2026-03-20

**Soundness:** 3
**Presentation:** 3
**Significance:** 3
**Originality:** 3
**Overall Recommendation:** 5
**Confidence:** 4

**Summary:**

This paper presents CausalX, a modular, "plug-and-play" framework designed to enhance multi-modal spatio-temporal forecasting by incorporating causal interpretability. It constructs a causal graph across different data modalities and time steps, refining it via a combination of multi-source causal constraints (Granger causality, do-calculus, TDMI, and VAE) and a prior-guided diffusion process. The model is architecture-agnostic, meaning it can be integrated into existing state-of-the-art backbones via feature fusion or graph augmentation. The authors show that CausalX can improve predictive accuracy and interpretability in tasks such as tropical cyclone and pedestrian trajectory forecasting.

**Compliance With Llm Reviewing Policy:**

Affirmed.

**Final Justification:**

My final recommendation for the paper is to Accept (5). The paper presents how a combination of causal losses can be integrated into a graph-based prediction setup, complementing existing state-of-the-art methods and leading to compelling performance improvements across two distinct scenarios. The rebuttal has raised my original score by clarifying the ease of transferring the learned graph structure across different prediction backbones.

**Key Questions For Authors:**

1. Do you have an explanation or intuition behind the performance drop in ZARA1 and ZARA2 on EigenTrajectory and SingularTrajectory, respectively? Both backbones have similar baseline performance but vastly different outcomes when CausalX is added.
2. To what extent do you expect the learned causal relations to depend on the employed backbone, e.g., considering the differences in the causal chord diagrams in Figure 5 (a) and (b)? Similarly, how easy is it to transfer the learned model from one backbone to another?
3. In Appendix A2, you remark, "In practice, the loss terms are combined by a simple unweighted sum, and we did not observe the need for additional loss-weight tuning". How did you come to that conclusion, and do you think this will be generally the case for more complex environments?

**Limitations:**

yes

**Strengths And Weaknesses:**

Strengths:
- CausalX is integrating multiple losses, such as Granger causality, Do-calculus, and TDMI, for incorporating causal reasoning into existing methods in a plug-and-play fashion, a critical research direction in forecasting.
- Performance improvements could be shown on multiple recent forecasting methods (MID, EigenTrajectory, and SingularTrajectory) in both physics-dominated and social-dominated environments (cyclone forecasting and pedestrian forecasting).
- The learned, causal representation as shown in the causal chord diagram (Figure 5) can lead to improvements in interpretability of the learned model and guide our understanding of a system beyond raw performance improvements.

Weaknesses:
- Table 1 shows that adding CausalX to recent forecasting methods is beneficial to performance on the selected benchmarks. Notably, prior work such as Trajectron++ and PCCSNET outperforms the chosen benchmarks (with and without CausalX) according to this table, raising the question of why they were not included in the comparison study. As far as I can tell, this choice is not discussed in the paper.
- Performance results are purely shown in terms of average performance; no standard deviation is indicated.
- It did not become clear to me how well CausalX scales to large numbers of variables, i.e., time and memory consumption, depending on the graph size and edge count. A deeper discussion of the time and space complexity of individual losses (e.g., impact on Granger compared to Do-calculus) could further improve the included ablation study (e.g., informing the decision of when to use all or only some of the losses in practice).
- Since causal ground-truth is unavailable, the claim that the proposed method has actually learned the causal relations in the tested scenarios remains unverified.

General remarks:
- Some figures could be improved in terms of legibility, e.g., showing fewer scenarios in Figure 3 but increasing line weight. Figure 6, specifically the shown graph, did not become entirely clear to me.
- The paper is overall very well written in terms of clarity and is easy to follow along, except for Section 3.1. Here, some rewriting for clarity may be beneficial to a reader who is not already deeply familiar with all topics covered.

---

> ### Author Rebuttal · Authors · 2026-03-29
>
> Thank you for recognizing the **importance of the problem and the empirical improvements** in our work, and for the helpful comments. The feedback is very helpful for improving the manuscript. Below are our responses.
>
> **Q1: Do you have an explanation or intuition behind the performance drop in ZARA1 and ZARA2 on EigenTrajectory and SingularTrajectory, respectively? Both backbones have similar baseline performance but vastly different outcomes when CausalX is added.**
>
> **A1:** Compared with more open scenes such as HOTEL or UNIV, **the ZARA scenes are more constrained by the environment**, including buildings, narrow walkable areas, and occasional vehicle interference, while these two backbones do not explicitly model all such factors. In particular, ZARA1 is a typical case where vehicles can visibly appear and affect pedestrian motion, so these scenes are not driven only by pedestrian-pedestrian interaction. Additionally, CausalX uses **only the information already used by the original backbone**. For a fair comparison, we should not manually add these environments that are not used by the original backbone. Thus, in ZARA scenes the trajectories are also strongly affected by scene constraints that are not fully captured by the backbone, so the improvement is less stable than in more open, interaction-dominated scenes.
>
> The different outcomes are related to the **different integration strategies**. In EigenTrajectory, CausalX is mainly integrated through **graph augmentation**, while in SingularTrajectory it is integrated through **feature fusion**. These different integration strategies make the two backbones organize and use information differently.
>
> **Q2: To what extent do you expect the learned causal relations to depend on the employed backbone (considering the differences in Fig. 5 (a) and (b))? Similarly, how easy is it to transfer the learned model from one backbone to another?**
>
> **A2:** We expect the learned graph to be **partially, but not entirely, backbone-dependent**. This can arise both from input differences across backbones and from their different representation and decoding preferences. Since CausalX is jointly trained with the backbone predictor rather than learning a graph in isolation, the learned relations naturally reflect both how the input information is organized and how the backbone internally encodes and uses it. However, the **overall relational picture remains stable across backbones** because of the same task objective under the same causal regularizers. For example, in Fig. 5(a) and (b), although the data inputs and some local rankings differ, both diagrams consistently highlight longitude, latitude, pressure, and wind speed as the main target-side factors.
>
> **Regarding transfer, CausalX remains easy to adapt across similar backbones**, while the required adaptation depends on whether the input is same. For backbones with different inputs, such as TC-Diffuser and TCN$_M$ in TC forecasting, adaptation is still lightweight and only requires adjusting the number of 1D auxiliary inputs $N_3$. Since CausalX constructs a graph over all input variables, a change in the input configuration naturally requires re-training. For similar tasks with aligned inputs, transfer is easier. For example, in pedestrian prediction, where different backbones use the same input, the graph-learning part of CausalX can be kept fixed, while the main adaptation is to re-train the fusion module that combines the learned graph / features with the target backbone.
>
> **Q3: How did you get the conclusion, "In practice, the loss terms are combined by a simple unweighted sum, and we did not observe the need for additional loss-weight tuning", and do you think this will be generally the case for more complex environments?**
>
> **A3:** This conclusion came from repeated experiments across **multiple backbones** and **two very different forecasting domains**, where the simple unweighted combination already produced reasonable and stable improvements. There was also a practical reason: if each backbone and dataset required another layer of loss-weight tuning, the usage complexity of CausalX would increase substantially, which would weaken its value as a general plug-and-play module.
>
> At the same time, we do not want to overstate this as a universal rule. The two settings studied here—**multi-agent pedestrian prediction** and **multi-modal tropical cyclone forecasting**—are **already complex**, and in our preliminary exploration on vehicle trajectory prediction, we did not observe that extra loss-weight tuning was indispensable. However, in more complex environments, **adaptive weighting or task-specific tuning may bring additional gains**, and we view that as a worthwhile direction for future work.
>
> *Due to space and time constraints, we may not include all additional results in current response, but we would be very happy to discuss these or any other questions during the author-reviewer discussion period.*

---

> > ### Author Rebuttal · Reviewer_b6KU · 2026-04-01
> >
> > The author's response has improved my understanding, especially concerning the transfer of the learned graph, and resolved my open questions regarding the method. I am raising my score accordingly to Accept (5).

---

> > > ### Author Response · Authors · 2026-04-01
> > >
> > > Thank you very much for your time, thoughtful comments, and positive update. We are delighted to know that our rebuttal was helpful and addressed your concerns, especially regarding the transferability of the learned graph. Your valuable suggestions have helped us further improve and enrich the paper. We sincerely appreciate your careful reading, constructive feedback, and support.

---

### Decision · Program_Chairs · 2026-04-30

**Decision:**

Accept (regular)

**Comment:**

This paper introduces a modular, plug-and-play framework for enhancing multi-modal spatio-temporal forecasting by incorporating causal interpretability. It constructs a causal graph across different data modalities and time steps, refining it via a combination of multi-source causal constraints and a prior-guided diffusion process. The experimental evaluation consists of a pedestrian trajectory prediction problem and tropical cyclone forecasting problem. The paper recieved 4 reviews with varying evaluations with the major points of contention being:

1. Dependence of the performance on the considered backbones EigenTrajectory and SingularTrajectory. AFter adding CausalX there were drops in some benchmarks and thus the oevrall efficacy of causaX was in question.

2. Lack of new causal insights and relevant structure of related work.

The rebuttal was detailed and did a great job at alleviating several concerns of the reviewers. At the end there wer 3 positive reviewers and 1 highly negative reviewer. During the discussion phase, a positive reviewer was willing to champion the paper. I do think that the points raised by reviewer aJix are valid and need addressing. For example, identifiability of causal effects is a critical issue and although I agree with the authors that this is a highly complex problem but in my opinion, without these guarantees, the overall story of the paper is far weaker. I thus recommend weak acceptance as of now.